# Polyphenol-stabilized coacervates for enzyme-triggered drug delivery

Wonjun Yim[1], Zhicheng Jin[2], Yu-Ci Chang [1], Carlos Brambila[2], Matthew N. Creyer[2], Chuxuan Ling[2], Tengyu He[1], Yi Li[2], Maurice Retout[2], William F. Penny[3], Jiajing Zhou [2] & Jesse V. Jokerst [1,2,4] ✉

Stability issues in membrane-free coacervates have been addressed with coating strategies, but these approaches often compromise the permeability of the coacervate. Here we report a facile approach to maintain both stability and permeability using tannic acid and then demonstrate the value of this approach in enzyme-triggered drug release. First, we develop size-tunable coacervates via self-assembly of heparin glycosaminoglycan with tyrosine and arginine-based peptides. A thrombin-recognition site within the peptide building block results in heparin release upon thrombin proteolysis. Notably, polyphenols are integrated within the nano-coacervates to improve stability in biofluids. Phenolic crosslinking at the liquid-liquid interface enables nano-coacervates to maintain exceptional structural integrity across various environments. We discover a pivotal polyphenol threshold for preserving enzymatic activity alongside enhanced stability. The disassembly rate of the nano-coacervates increases as a function of thrombin activity, thus preventing a coagulation cascade. This polyphenol-based approach not only improves stability but also opens the way for applications in biomedicine, protease sensing, and bio-responsive drug delivery.

Coacervate droplets are cell-like compartments consisting of a condensed solution of macromolecules such as peptides[1], copolymers[2], RNAs[3], and a diluted phase of the remaining liquid[4]. They formed instantly via liquid-liquid phase separation driven by non-covalent interactions such as electrostatic[5], hydrophobic[6], and hydrogen bonding[7]. The rich concentrations of biomolecules within the coacervates often mimic the physicochemical environments of living cells. Thus, coacervates have been extensively studied to understand the early stage of cell evolution[8–10]. The physical properties of coacervate droplets rely on the structural- and chemical- properties of their constituent building blocks[6]. However, the lack of membranes often leads to a rapid coalescence or collapse of the coacervate phases (i.e., poor stability)[11,12]. The absence of a physical membrane also limits their ability to mimic the selective permeability of cellular membranes[13,14]. These limitations challenge the promise of coacervate droplets for protocell models[14], biomedicine[15], drug delivery[16], and biosensing[17] applications.

To address this stability issue, researchers have largely focused on developing hybrid protocell models consisting of coacervate-based interiors surrounded by membranes such as terpolymer[2], phospholipids[18], erythrocyte[19], and polysaccharide[20] layers. These surrounding membranes either coat or encapsulate the coacervate droplet, which in turn enhances its stability. In the coating approach, a coacervate droplet is used as a template: Two opposite-charged building blocks initially form complex coacervates followed by the in-situ formation of membranes[18–21]. Alternatively, encapsulation of coacervates within liposomes is based on microfluidic techniques. Coacervates encased within liposomes have shown great potential in the development of a bio-responsive platform capable of reacting to changes in pH[22], osmotic gradient[23], and temperature[24]. However,

[1]Materials Science and Engineering Program, University of California San Diego, La Jolla, CA, USA. [2]Aiiso Yufeng Li Family Department of Chemical and NanoEngineering, University of California San Diego, La Jolla, CA, USA. [3]Division of Cardiology, VA San Diego Healthcare System, University of California San Diego, La Jolla, CA, USA. [4]Department of Radiology, University of California San Diego, La Jolla, CA, USA. ✉e-mail: jjokerst@ucsd.edu

these strategies often result in limited permeability of the surrounding membrane, which can hinder the penetration and/or release of large biomolecules[22,25].

Our goal here is to establish a stable and enzyme-responsive coacervate platform. We first engineer a nano-sized coacervate made of bio-inspired peptides and the anticoagulant heparin. Heparin plays an important role in surgical and cardiovascular medicine due to its short half-life, reversible nature, and low cost[26]. However, heparin is difficult to manage and requires blood draws and central labs[27]; therefore, the controlled release of heparin via the enzymatic activity of clotting factors is gaining interest[28–32]. Living organisms maintain hemostasis through precise molecular feedback regulations[33]. For example, vascular injury triggers a coagulation cascade process where clotting factors activate prothrombin to thrombin, transforming fibrinogen into insoluble fibrin by cleavage. Together with platelet activation, this process produces stable fibrin clots to prevent excessive bleeding[33]. We envision that by incorporating a feedback loop system within the coacervates, they could regulate heparin release based on thrombin activity. Increasing environmental thrombin levels would promptly trigger heparin release, while normal physiological thrombin levels would leave the coacervates intact—thus, there would be no risk of excessive bleeding[34,35].

This work thus incorporates a thrombin cleavage site within the peptide used to make the nano-coacervate, resulting in the release of heparin as a function of concentration-dependent thrombin proteolysis. Importantly, we demonstrate enhanced coacervate stability via polyphenol-mediated supramolecular networks while maintaining their thrombin proteolytic activity. This structural and colloidal enhancement is obvious, clearly visualized by transmission electron microscopy (TEM)—they had exceptional stability in challenging conditions and various biofluids but could still specifically release the heparin cargo. The disassembly rate of nano-coacervates rapidly increased in response to thrombin proteolysis in human plasma. Overall, our approach of utilizing polyphenols to stabilize coacervates and preserve the bioactivity for enzymatic degradation is a simple yet powerful tool in the fields of biomedicine, biosensing, and enzyme-triggered drug delivery.

## Results

### Nano-coacervates driven by a tyrosine and arginine peptide

The Mytilus edulis foot protein 5 (Mefp-5) in mussels contains repetitive DOPA and lysine (K) groups that provide positively charged residues with hydrophobic interactions[36]. This enables Mefp-5 to interact with a wide array of materials through either covalent or noncovalent interactions (Supplementary Fig. 1)[37–39]. The first step of designing our system was to determine whether a short peptide composed of tyrosine (Y) and arginine (R) could form a coacervate droplet with heparin (average $M_w$: 15,000 Da) (Fig. 1a). Heparin is a glycosaminoglycan with repeating sulfate units that provide negative charge and a polysaccharide structure for efficient binding with antithrombin[40]. Our previous studies revealed that heparin could assemble with small molecular dyes via strong electrostatic and hydrophobic interactions[29,41] suggesting that the repetitive YR sequence might also readily trigger the formation of coacervates. To test our hypothesis, we synthesized a short YRYR peptide (referred to as C2) and mixed C2 (0.05–1.5 mM) with heparin (50 U/ml). Upon interaction with heparin, the C2 peptide instantly formed coacervate droplets of varying sizes confirmed by dynamic light scattering (DLS) (Fig. 1b and Supplementary Fig. 2). Micro-sized coacervates exhibited a broad extinction spectrum, likely due to increased light scattering while the light absorption of nano-sized coacervates increased at more blue-shifted wavelengths (Fig. 1c and Supplementary Fig. 3).

To further determine how many repeating YR units are needed to induce coacervation, YR, YRYR, and YRYRYRYR peptides were synthesized (referred to as C1, C2, and C3, respectively) as confirmed by

matrix-assisted laser desorption/ionization (MALDI-TOF) (Supplementary Fig. 4). At a constant heparin concentration of 50 U/ml, C2, and C3 peptides formed coacervate droplets of different sizes from 70 nm to 1 μm while the C1 peptide was incapable of forming coacervates, indicating that heparin-based coacervation requires at least two YR units (Fig. 1d and Supplementary Figs 5, 6). We further added six glycine (G) between YR sequences (YRG$_6$YR, referred to as C4) to confirm the impact of charge density and peptide length. The results showed that both C2 and C4 require comparable peptide concentrations for coacervation, suggesting that the number of YR units (i.e., valence charge) plays a key role in coacervate formations rather than the steric bulk (extra glycine units) (Supplementary Fig. 7).

In addition, various heparin concentrations from 0.25 to 50 U/ml were combined with a constant peptide concentration of 1 mM, confirming that the coacervation relies on the number of YR units and heparin concentration (Fig. 1e and Supplementary Fig. 8). The formation of coacervate droplets led to an increase in turbidity, thus changing the color from transparent to white (Fig. 1f). We also observed that the strong interactions between heparin and C2 peptide resulted in high loading efficiency of nano-coacervates (99.5–100%) (Fig. 1g and Supplementary Fig. 9). The nano-coacervates limit their growth and maintain their size and phase separation even under the centrifugation of 7 × g: There was no coalescence or merging.

To confirm the interactions governing the coacervate formation of the C2–heparin complex, nano-coacervates were incubated with PEG2000, citric acid, urea, Triton X-100, SDS, DMF, and DMSO, respectively. Triton X-100 and SDS can break non-ionic or ionic interactions; DMSO and DMF are organic solvents that can destroy pi-pi interaction[42]; and urea can break the hydrogen bonding[43]. The nano-coacervates were disassembled in urea, Triton X-100, SDS, DMF, and DMSO conditions, indicating that electrostatic, pi-pi interaction, and hydrogen bonding were involved in the formation of C2–heparin coacervates (Fig. 1h and Supplementary Fig. 10). Nano-coacervates were stable at low pH (1–5), but they disassembled at high pH (over 9) due to the deprotonation of the guanidine group (Fig. 1i). The isoelectric point of arginine is 10.8[44]. Finally, micro- and nano-coacervates were visually observed using multiple wavelength nanoparticle tracking analysis (M-NTA)[45] that further verified the narrow size distribution of nano-coacervates from DLS data (Fig. 1j, and Supplementary Fig. 6, and Supplementary Movies 1–3).

### Enzyme-responsive coacervate droplets

Thrombin is a central enzyme in hemostasis and activates the fibrin network and platelets for blood clots when damaged tissue triggers factor VII (details described in Supplementary information)[46,47]. Heparin can prevent these clotting cascades because it contains saccharide units that bind to antithrombin, inactivating a number of coagulation enzymes[48]. We envisioned that if thrombin can cleave the peptide building block and disassemble the coacervates, then the coacervates could be an enzyme-responsive platform that can release the encapsulated heparin in response to thrombin proteolysis (Fig. 2a). To achieve this, we added a thrombin cleavage site (LVPR ↓ GS)[49] in the middle of the C2 sequence: YRLVPRGSYR (referred to as C5) (Table 1). The thrombin proteolysis would result in fragment peptides that only contain one YR unit which is not sufficient for phase separation as depicted in Fig. 1d. Initially, we confirmed that the C5 peptide could form nano-coacervates with heparin, leading to an increase in turbidity (Fig. 1d and Supplementary Fig. 7).

The nano-coacervates were then incubated with various concentrations of thrombin from 0.05 to 2.5 μM. The turbidity of the nano-coacervates decreased due to thrombin proteolysis with higher concentrations of thrombin leading to rapid dissociation of the nano-coacervates (Fig. 2b and Supplementary Fig. 11). Notably, nano-coacervates composed of the scramble sequence (i.e., C6) showed negligible change in turbidity before and after thrombin incubation

(Fig. 2c and Supplementary Fig. 12). We further confirmed the mass peak of the fragment peptide (845.63, YRLVPR) after thrombin cleavage via MALDI-TOF (Fig. 2d).

In addition, the photoluminescence (PL) performance of C7-encapsulated nano-coacervates was examined upon thrombin proteolysis. We conjugated a sulfo-Cy5.5 dye with C5 peptide using an amine-NHS coupling (i.e., C7) and encapsulated C7 peptides within the nano-coacervates (details described in Supplementary Fig. 13). After C7 encapsulation, the PL signal of the C7 peptide was quenched, and the nano-coacervates exhibited a red-shifted absorption peak at 688 nm. This shift was likely due to increased intermolecular interactions, such as pi-pi stacking between tyrosines[50], as well as electrostatic interactions between heparin and lysine within the nano-coacervates.

Thrombin cleavage released sulfo-Cy5.5, recovering an absorption peak at 676 nm and its PL intensity at 700 nm (Fig. 2e). The kinetics of PL activation increased as a function of thrombin concentration: Higher concentrations of thrombin led to a more rapid disassembly of nano-coacervates, promptly activating the PL signal (Fig. 2f). Furthermore we determined the specificity constant ($k_{cat}/K_M$) by thrombin using a fluorogenic substrate (Cy5.5−YRLVPRGSYRC−Cy3, referred to as C8) (Fig. 2g and Supplementary Figs 14–15) and was 0.91 $\mu M^{-1}s^{-1}$, which is as fast as the thrombin-catalyzed conversion of human fibrinogen to fibrin (1.88 $\mu M^{-1}s^{-1}$)[51]. The specificity of our system was further tested toward other proteins such as bovine serum albumin (BSA), hemoglobin (Hemo), SARS-CoV-2 main protease (M^pro), and α-amylase at the same enzyme concentration of 5 μM. No PL signal was activated

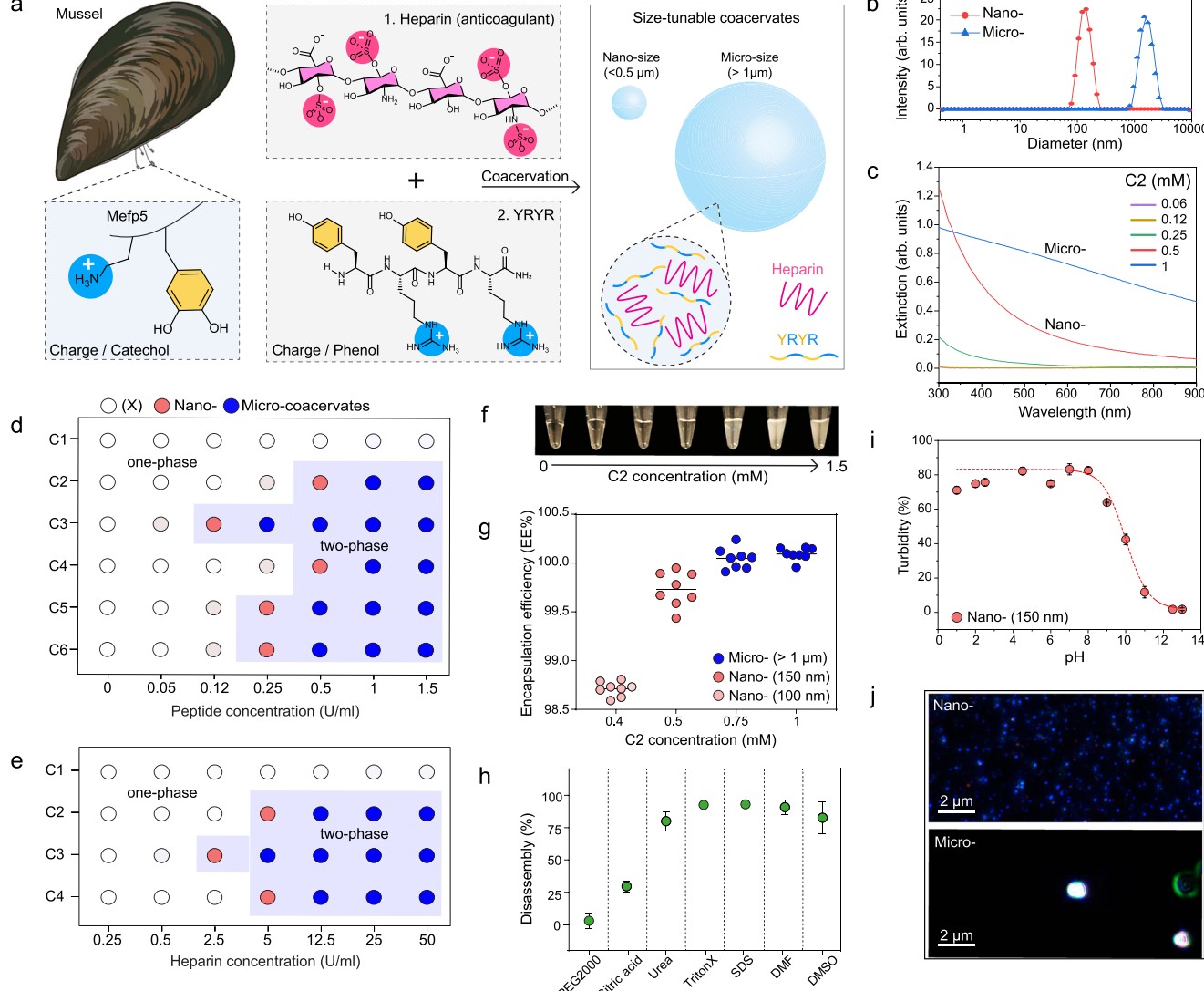

**Fig. 1 | Size-tunable coacervate driven by YR-heparin interactions. a** Schematic illustration of coacervate design using heparin and a YR-based short peptide. **b** DLS data of C2-heparin coacervates, showing their nano (198 ± 3.3 nm) or micro (>1 μm) sizes. **c** UV-vis spectra of nano- and micro-coacervates. Formation of coacervates as a function of different peptide (**d**) and heparin (**e**) concentrations. Six different peptide sequences (details described in Table 1) are examined to study the impact of the charge, concentration, number, thrombin recognition site, and length of YR-based peptides for heparin coacervation. The blue area indicates coacervate formation (i.e., phase separation). Red and blue dots indicate nano- and micro-sized coacervate formation, while empty dots represent no coacervate formation. **f** The photograph shows the increased turbidity as a function of coacervate formation.

**g** High encapsulation efficiency of nano- and micro-coacervates. Eight dots indicate the encapsulation efficiency of eight independent coacervate samples. Stability test of nano-coacervates in different conditions (**h**) including PEG2000, citric acid, urea, Triton X-100, SDS, DMF, and DMSO, and different pH (**i**). **j** M-NTA images of nano- and micro- coacervates. Small blue dots represent monodispersed nano-coacervates, and large white dots indicate scattered micro-coacervates. The experiment was repeated three times independently with similar results. Data in (**h**) and (**i**) represent mean ± SD (n = 3). Figure 1/panel (**a**) Created with BioRender.com released under a Creative Commons Attribution-NonCommercial-NoDerivs 4.0 International license.

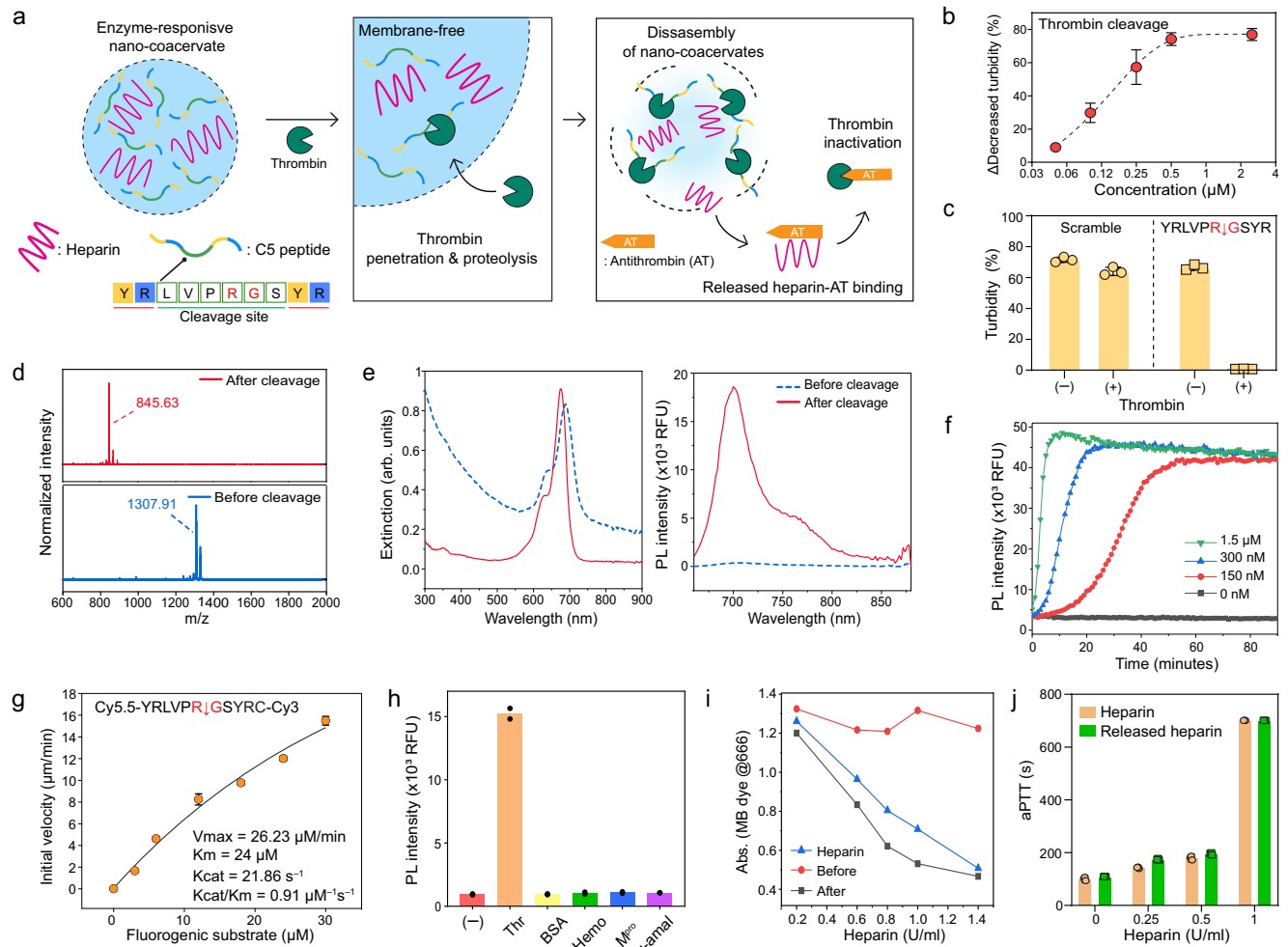

**Fig. 2 | Thrombin-responsive nano-coacervates. a** Schematic illustration of heparin release through disassembly of nano-coacervates driven by thrombin proteolysis. The released heparin binds antithrombin which induces thrombin inactivation. **b** Decreased turbidity of nano-coacervates as a function of thrombin concentrations. **c** Turbidity changes of C5- and C6-based nano-coacervates with and without thrombin (5 μM). **d** MALDI-MOF data before and after thrombin cleavage, confirming the mass peak of the parent ($M_w$: 1307.91) and its fragment ($M_w$: 845.63). The N-terminus of C5 was acetylated. **e** UV-vis and PL spectra of C7-encapsulated nano-coacervates before and after thrombin cleavage. The quenched PL signal of sulfo-Cy5.5 dyes activated as a function of nano-coacervates' disassembly. **f** Time-dependent $PL_{670\,nm}$ changes driven by thrombin cleavage. **g** $k_{cat}/K_M$ determination for C5 peptide cleavage driven by thrombin proteolysis. The thrombin (20 nM) was incubated with a fluorogenic substrate ([S]$_0$ = 0–30 μM,

sequence shown on top in the panel box), and the product concentration at 30 min was used. Data was fit to the Michaelis–Menten equation (see Supplementary information 2.8). **h** Specificity test using different biological proteins including thrombin (Thr), bovine serum albumin (BSA), hemoglobin (Hemo), main protease of SARS-CoV-19 ($M^{pro}$), and α-amylase (50 U/ml). A sample without any proteins is referred to as a negative control. **i** Decreased absorbance of MB dye before and after the addition of thrombin. The disassembly of nano-coacervates released heparin, leading to a decrease in the absorbance of MB dye while intact nano-coacervates showed a negligible change in absorbance (Supplementary Fig. 17). **j** aPTT test of heparin and released heparin from the disassembly of nano-coacervates. Data in (**f**) and (**h**) represent the mean value of two independent samples. Data in (**b**), (**c**), (**g**), and (**j**) represent mean ± SD (*n* = 3).

in the presence of BSA, Hemo, and other enzymes (Fig. 2h and Supplementary Fig. 16).

Finally, a methylene blue (MB) assay was used to confirm the released heparin from the disassembly of nano-coacervates[52]. Heparin (from 0.125 to 5 U/ml) had a linear decrease in the absorption peak of the MB dye at 666 nm, subsequently causing a redshift of the peak to 566 nm due to the formation of heparin-MB complex (Supplementary Fig. 9). After incubation with thrombin, the disassembled samples were centrifuged to collect the supernatant. The supernatant obtained from the disassembled nano-coacervates decreased the absorption peak of the MB dye to 666 nm. Conversely, nano-coacervates without thrombin exhibited negligible changes in absorption, thus indicating that intact nano-coacervates did not release heparin (Fig. 2i and Supplementary Fig. 17). In addition, the released heparin could prevent plasma coagulation as confirmed by an activated partial

thromboplastin time (aPTT) test (Fig. 2j). The C5 peptide only showed plasma coagulation due to lack of anticoagulant ability (Supplementary Fig. 18). Collectively, our coacervate-based heparin delivery offers an enzyme-responsive mechanism capable of releasing heparin in response to thrombin proteolysis for controlled anticoagulant therapy.

## Polyphenol-stabilized nano-coacervates
A major drawback of using coacervate is their limited stability in biofluids. Human plasma, which contains diverse proteins, clotting factors, and ions, readily disrupts coacervate phases (Supplementary Fig. 19). We hypothesized that tannic acid (TA) could enhance colloidal and structural stability because multiple catechol groups in TA could create a strong supramolecular network with tyrosine[53] and polysaccharide[54] which are major structural components in the

**Table 1 | Description of peptide building blocks for nano-coacervates**

| Peptide Name | Peptide sequence Am: amide (CONH2) | Net Charge | M.W. (g mol$^{-1}$) | Description |
|---|---|---|---|---|
| C1 | NH2 –YR–Am | +2 | 337.3 | A single YR |
| C2 | NH2 –YRYR–Am | +3 | 655.4 | Two repetitive YR |
| C3 | NH2 –YRYRYRYR–Am | +5 | 1293.7 | Four repetitive YR |
| C4 | NH2 –YRGGGGGGYR–Am | +3 | 997.5 | Six glycine between YR unit |
| C5 | NH2 –YRLVPRGSYR–Am | +4 | 1264.7 | Addition of thrombin cleavage sequence |
| C6 | NH2 –YRSLRGPVYR–Am | +4 | 1264.7 | Addition of scramble sequence |
| C7 | Sulfo-Cy5.5 –YRLVPRGSYR–Am | +3 | 2151.1 | Sulfo-Cy5.5 dye conjugation |
| C8 | Cy5.5 –YRLVPRGSYRC(Cy3)–Am | +3 | 2512.7 | Fluorogenic substrate for Kcat/KM test |

nano-coacervates. To verify this, the nano-coacervates were encapsulated with TA molecules of 0.05, 0.25, 0.5 mM under pH 8.5 (referred to as NC-TA$_{0.05}$, NC-TA$_{0.25}$, and NC-TA$_{0.5}$, respectively) (Fig. 3a). These NC-TAs showed narrow size distributions (polydisperse index (PDI) ≤ 0.1), and similar hydrodynamic diameters (Fig. 3b). The average diameter of each NC-TAs was 253.7 ± 9.4 nm (NC-TA$_0$), 221.8 ± 4.8 nm (NC-TA$_{0.05}$), 228.4 ± 7.1 nm (NC-TA$_{0.25}$), and 276 ± 4.3 nm (NC-TA$_{0.5}$), respectively. TA encapsulation resulted in a notable increase in the extinction value of NC-TAs in the near ultraviolet (UV) region, and the color of the sample changed from white to yellowish-brown (Fig. 3c and Supplementary Fig. 20). Fourier-transform infrared spectroscopy (FTIR) data evidenced the TA encapsulation as the appearance of C-O vibration (1320 cm$^{-1}$) and 1, 3-disubstituted benzene rings around (1100−700 cm$^{-1}$) (Fig. 3d)[55]. To further understand TA encapsulation within the coacervates, NC-TAs were incubated at different pH and solvent conditions. NC-TAs remained stable until pH 10 but disassembled beyond 11 due to deprotonation (Fig. 3e). In addition, DMF, DMSO, and SDS led to the disassembly of NC-TAs, suggesting that electrostatic and pi-pi interactions were involved in TA encapsulation (Fig. 3f). NC-TAs showed higher stability in pH 9 and under urea compared to pristine nano-coacervates.

We then attempted to image nano-coacervates before and after the TA encapsulation using TEM and scanning electron microscopy (SEM). NC-TAs maintained their size and spherical shape even in the vacuum condition confirmed by both TEM and SEM (Fig. 3g and Supplementary Fig. 21) while the nano-coacervates without TA collapsed and deformed during the drying process (Supplementary Fig. 22). Furthermore, we used tomography imaging at various angles ranging from −30° to 60° to illustrate the interface between the bottom of NC-TA$_{0.05}$ and the underlying substrate (i.e., TEM grid). Fig. 3h clearly shows the height of a single NC-TA$_{0.05}$ at 60°, indicating that TA molecules formed a rigid supramolecular network and enhanced the structural integrity of the nano-coacervates (Supplementary Fig. 23). High-angle annular dark field (HAADF) and energy-dispersive X-ray spectroscopy (EDX) were also utilized to confirm the elemental components of NC-TAs (Figs. 3i–j). EDX mapping revealed that C, N, O, and S signals were observed in a single NC-TA$_{0.05}$ which are components of TA, C5 peptide, and heparin (Fig. 3j and Supplementary Fig. 24).

We also discovered that polyphenol encapsulation can be applied to micro-sized coacervates (MC-TAs). The optical image illustrates the uniformly dispersed micro-coacervates after TA encapsulation (Supplementary Fig. 25). The spherical shapes and sizes of the dried MC-TA$_{0.05}$ were confirmed by the SEM technique (Fig. 3k and Supplementary Fig. 26). Notably, we observed significantly improved colloidal stability under the centrifugation of $3 \times g$. The coacervate droplets without TA showed a 98.7% decrease in turbidity while MC-TA$_{0.05}$ decreased by only 1.4% (Supplementary Fig. 27). Lastly, coumarin boronic acid was conjugated with TA for confocal imaging to verify TA encapsulation within coacervate droplets. HPLC was utilized to remove free coumarin dyes from TA-coumarin before encapsulation (Supplementary Fig. 28). Fig. 3l shows a uniformly distributed fluorescent signal of TA-coumarin conjugates from inside the MC-TAs. This result indicates that TA is encapsulated within the coacervates rather than being membrane-coated[10,56].

## Preserving enzymatic activity of nano-coacervates with enhanced stability

The formation of polyphenol networks within the coacervates significantly enhances stability; however, highly constructed supramolecular networks could adversely affect the proteolytic efficiency[6] (Fig. 4a). To examine this, nano-coacervates encapsulated with various TA concentrations were incubated in NaCl for 1 h. We observed a less decrease in turbidity (T) as a function of increased TA encapsulation while nano-coacervates without TA were dissociated within the 30 s (Fig. 4b): Turbidity (T$_{after}$/T$_{before}$) of NC-TA$_0$, NC-TA$_{0.17}$, NC-TA$_{0.33}$, and NC-TA$_1$ were 7%, 36%, 53%, and 92%, respectively. In contrast, high TA encapsulation led to a decrease in proteolytic activity. The turbidity changes of NC-TAs were measured after incubating different concentrations of alpha-thrombin (M$_w$: 37.4 kDa) ranging from 0.06 to 1 μM. NC-TA$_{0.5}$ showed a reduced decrease in turbidity compared to NC-TA$_{0.25}$ when incubated with the same concentration of thrombin (Fig. 4c and Supplementary Fig. 29). NC-TA$_1$ exhibited negligible changes in turbidity, indicating that the excessive TA encapsulation could prevent thrombin-driven coacervate disassembly.

After identifying a critical TA encapsulation point for preserving thrombin proteolytic activity, we examined the colloidal stability of NC-TA$_{0.13}$ under various biological environments. The NC-TA$_{0.13}$ exhibited high colloidal stability in glutamine, glucose (5.6 mM), human albumin (0.6 mM), DPBS, NaOH (pH 10), 60 C°, NaCl (150 mM), fibrinogen (8.8 μM), 50% of Dulbecco's Modified Eagle Medium (DMEM), serum, saliva, and urine (Fig. 4d and Supplementary Figs. 30–31). Both pristine nano-coacervates and NC-TAs containing either C7 peptides or heparin-FITC were incubated in 50% human plasma, respectively to examine improved stability. The quenched fluorescence of the C7 peptide and heparin-FITC was activated as a function of the disassembly of nano-coacervate (Fig. 4e). NC-TA$_{0.13}$ exhibited a 3.3-fold decrease in the PL activation rate of C7 peptide than NC-TA$_0$, indicating enhanced stability in human plasma. Simultaneously, thrombin could accelerate heparin release from NC-TA$_{0.13}$ (Fig. 4f). The disassembly rate of NC-TA$_{0.13}$ increased by 4.2-fold upon the addition of thrombin (500 nM) which falls within the physiologic range of free thrombin concentration. Physiologic concentrations of free thrombin during coagulation reactions range over 500 nM[57]. We also monitored this disassembly process using heparin-FITC (Supplementary Fig. 32). NC-TAs exhibited a decrease in PL activation of heparin-FITC compared to pristine nano-coacervates in human plasma. Concurrently, the addition of thrombin rapidly increased the PL activation rate of heparin-FITC by 1.8-fold, recovering PL intensity within 10 min (Fig. 4g). There was no fluorescence quenching of C7 peptides and heparin-FITC either by the background medium (i.e., human plasma) or by TA molecules (Supplementary Fig. 33). The release kinetics of heparin are different than the peptide because the

concentration of coacervate samples and fluorescent dye-conjugates (C7 peptide and heparin-FITC) were different; the ratio could be tuned to control kinetics. We next examined the cell viability and cellular reactive oxygen species (ROS) levels of NC-TA$_{0.13}$ using human

umbilical vein endothelial cells (HUVECs), respectively. NC-TAs and their structural components such as C5 peptide, heparin, and TA showed high cell viability (>83%) and minimal ROS intensities (Fig. 4h). NC-TA$_{0.13}$ also exhibited low cytotoxicity against human embryonic

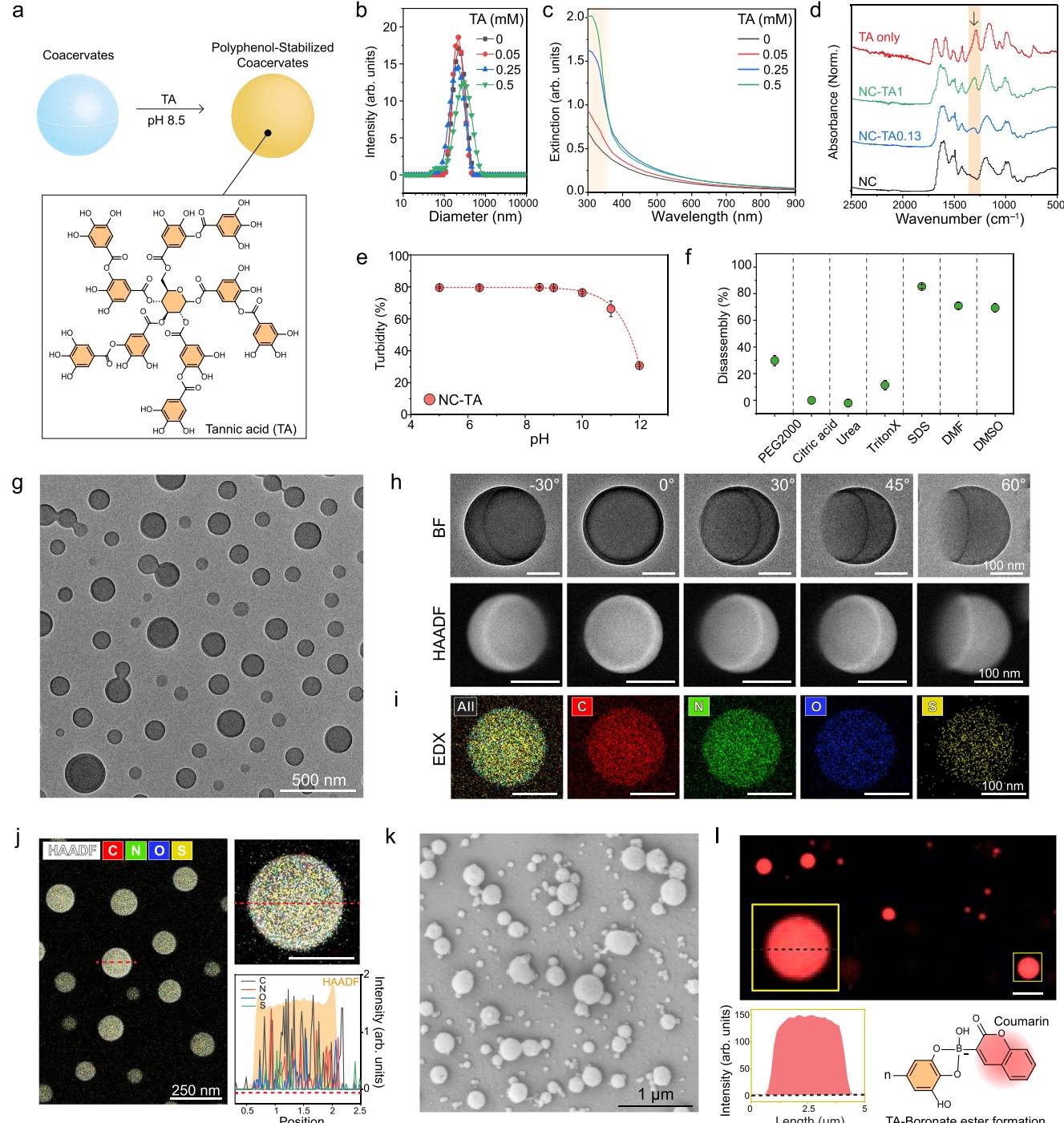

**Fig. 3 | Material characterizations of polyphenols-encapsulated coacervates. a** Schematic illustration of TA encapsulation within the coacervates. DLS (**b**) UV-vis spectra (**c**) and FTIR (**d**) of NC-TAs. The brown region in (**c**) and (**d**) indicates the appeared peaks after TA encapsulation. Stability test of NC-TAs in different pH (**e**) and different conditions (**f**) including PEG2000, citric acid, urea, Triton X-100, SDS, DMF, and DMSO. **g** TEM image of NC-TA$_{0.13}$. **h** Bright field (BF) and HAADF images of a single NC-TA$_{0.13}$ at different angles. Supplementary Figs 22–24 show multiple NC-TA$_{0.13}$ at different angles with low magnification. **i,j** EDX elemental mapping of a single NC-TA$_{0.13}$, showing C, N, O, and S elements which are major components of heparin, peptide, and TA. The red-dotted line indicates the region used for the EDX

mapping. The scale bar in (**h**–**j**) represents 100 nm. **k** SEM of micro-coacervates (i.e., MC-TAs). **l** Confocal image of MC-TAs encapsulating TA-coumarin conjugates. The yellow box indicates a single MC-TA with high magnification that highlights the evenly distributed fluorescent signal of TA-coumarin inside the MC-TA. This result reveals that TA is encapsulated within the coacervates. The scale bar represents 5 μm. Coumarin boronic acid was linked with hydroxyl groups in TA, forming a boronate ester, and the conjugates were purified using HPLC before encapsulation (Supplementary Fig. 28). Data in (**e**) and (**f**) represent mean ± SD (*n* = 3). The experiment in (**g**–**l**) was repeated three times independently with similar results.

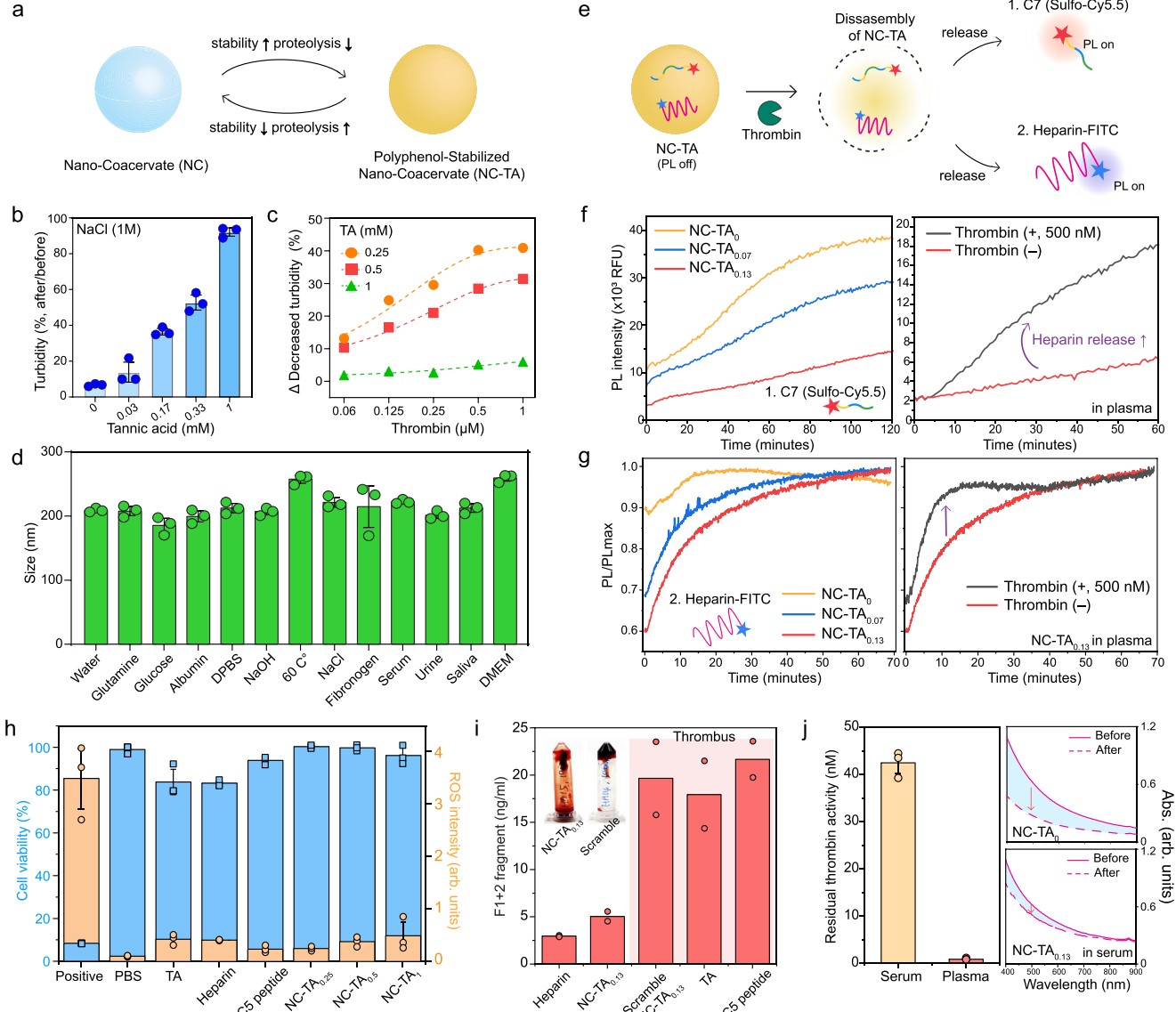

**Fig. 4 | Enhanced stability of NC-TAs and their proteolytic efficiency.**
**a** Schematic illustration of a trade-off between stability and proteolysis-based disassembly of NC-TAs. NC-TAs increased stability (**b**) in NaCl as a function of TA encapsulations while reducing their proteolytic efficiencies (**c**). Thrombin was unable to dissociate NC-TA$_1$. **d** Size profiles of NC-TA$_{0.13}$ in different biological environments. **e** Schematic illustration of monitoring either C7 peptide (**f**) or heparin-FITC (**g**) during disassembly of NC-TAs by thrombin. The left panels in (**f**, **g**) show a decrease in the PL activation rate of NC-TAs compared to nano-coacervates (i.e., NC-TA$_0$) due to improved stability in 50% human plasma. The right panels in (**f**, **g**) illustrate the addition of thrombin rapidly activates the PL intensity of C7 peptide or heparin-FITC, indicating proteolysis-driven heparin release. **h** Cell viability (blue) and ROS intensity (orange) of HUVEC incubating with PBS, TA, heparin,

C5 peptide, and NC-TAs, respectively. **i** Prothrombin F1 + 2 peptide concentrations of NC-TA$_{0.13}$ TA, C5 peptide, and NC-TA$_{0.13}$ made of scramble peptide (i.e., C6) from whole human blood incubation. The inserted photo shows a strong blood clot from blood anticoagulation from NC-TA$_{0.13}$ (left) and scramble NC-TA$_{0.13}$ (right).
**j** Residual thrombin activity in human serum and plasma. Human serum shows higher residual thrombin activity comparable to 42.5 nM of alpha-thrombin. The graphs on the right panel in (**j**) represent absorbance changes of nano-coacervates and NC-TA$_{0.13}$ before and after 1 h incubation in 50% human serum, showing higher stability of NC-TA$_{0.13}$ than pristine nano-coacervates. Data in (**c**), (**f**), (**g**), and (**i**) represent the mean value of two independent samples. Data in (**b**), (**d**), (**h**), and (**j**) represent mean ± SD (*n* = 3).

kidney (HEK) 293 cells. NC-TAs led to minimal red fluorescence of propidium iodide (PI), which corresponds to cell viability of 95% (Supplementary Fig. 34).

We tested the anticoagulant performance of NC-TA$_{0.13}$ in whole human blood using a human prothrombin fragment 1 + 2 (F1 + 2) enzyme-linked immunosorbent assay (ELISA) kit. Fresh whole blood was collected using an EDTA-treated blood collection tube. Free heparin (0.6 U/ml), C6 peptide, TA, NC-TA$_{0.13}$, and scramble NC-TA$_{0.13}$ were incubated with whole human blood at the same concentration, respectively. Calcium chloride was then used to trigger blood clot formation. We observed a negligible difference in F1 + 2 formation

between heparin and NC-TA$_{0.13}$, confirming active blood anticoagulation driven by heparin released from NC-TA$_{0.13}$ (Fig. 4i and Supplementary Figs. 35, 36). In contrast, a strong thrombus was observed in TA, C5 peptide only, and scramble NC-TA$_{0.13}$. Note that scramble NC-TA$_{0.13}$ was comprised of a scramble C6 peptide, serving as a non-responsive control. Lastly, we examined the residual thrombin activity of human serum and plasma using a thrombin chromogenic substrate. Human serum exhibited at least 20-fold higher residual thrombin activity than plasma (Fig. 4j). The residual thrombin activity in human serum was comparable to that of 42.5 nM of the alpha-thrombin (Supplementary Fig. 37). This difference arises

because human plasma is obtained by anticoagulating blood, which prevents the clotting cascade, thus inhibiting thrombin formation. In contrast, human serum is collected by allowing blood to clot, during which thrombin is generated from prothrombin, resulting in higher residual thrombin activity. Nano-coacervates without TA showed a 56% decrease in absorbance at 500 nm due to residual thrombin activity while NC-TA$_{0.13}$ showed only an 18% decrease in 50% human serum, indicating enhanced stability through polyphenol encapsulations (Fig. 4j and Supplementary Fig. 37).

## Discussion

In summary, we report the self-assembly of YR-based peptides with heparin, forming coacervates through a combination of electrostatic, hydrogen bonding, and hydrophobic interactions. This assembly can produce a range of sizes from nano to micro-coacervates. In addition, the peptide building blocks involve a thrombin recognition site to incorporate a hemostasis feedback loop system within the coacervate for controlled heparin release. Increasing thrombin levels trigger the disassembly of the coacervates, rapidly releasing heparin, while the absence of thrombin leaves the coacervates intact.

The nano-coacervate was further stabilized via a polyphenol-mediated supramolecular network to improve its stability in human plasma. TA encapsulation improves the structural integrity of nano-coacervates as clearly visualized by TEM. Importantly, we demonstrated a critical TA concentration for preserving both thrombin proteolytic activity and colloidal stability. NC-TAs exhibited high stability under various biological conditions. Simultaneously, the disassembly rate of NC-TAs rapidly increased upon the addition of thrombin, leading to heparin release in human plasma. NC-TAs also feature bioresponsive anticoagulant performance in the whole human blood and high biocompatibility with HUVEC and HEK293 cells.

Coacervates containing catechol as structural building blocks have shown significant potential in drug delivery systems, particularly for gastrointestinal diseases[58] due to their strong adhesiveness[59] capable of prolonged retention in the gastrointestinal tract. Our nano-coacervates strengthened by polyphenols also showed superior coating ability on inert substrates and maintained high stability in whole human blood (Supplementary Fig. 38). Future work will incorporate coacervates on medical devices such as a drug-eluting stent for on-demand anticoagulant delivery. Studies on inflammatory aspects such as plasma viscosity, procalcitonin, and C-reactive protein levels–as well as the elimination of particles from the circulation by phagocyte update or clearance in the kidney, spleen, and liver–are needed to validate their value in translational nanomedicine. Taken together, our polyphenol-based platform to stabilize coacervates and preserve bioactivity may have a scope well beyond drug delivery, extending its applications to biomedicine, protease sensing, and hybrid protocell models.

## Methods

A human blood specimen was collected from one male subject under approval from the institutional review board (IRB) of UC San Diego and the VA San Diego (#H170005). All subjects gave written informed consent. All work was done in accordance with the Declaration of Helsinki. We did not investigate sex as a biological variable because we are unaware of sex-based differences in thrombosis.

### Experimental Details

**Nano-Coacervates Preparation.** Briefly, 4 mg of the C5 peptides was dissolved in 3 ml of deionized water. Subsequently, 300 μL of heparin solution (100 U/ml) was mixed with 200- or 300- μL of the C5 peptides, immediately formulating nano- or micro-sized coacervates. The heparin concentration required to form coacervates depends on peptide concentration and the number of repetitive YR units in the peptide building block. The color of the solution became turbid once coacervation occurred. The size of coacervate droplets depends on the

concentrations of the heparin and C5 peptides used for the coacervation. For example, 0.25 mM of the C5 peptide formed nano-coacervates at a constant heparin concentration of 50 U/ml while 0.5 mM of the C5 peptide formed micro-coacervates. The resulting product was purified by centrifugation at $3 \times g$ for 10 min to remove any unreacted heparin or peptides. The pellet containing nano-coacervates was re-dispersed in MQ water for future use. However, the micro-sized coacervates were deformed, and the coacervate phase disappeared after centrifugation at $3 \times g$. All peptide sequences were synthesized using an AAPPTEC peptide synthesizer (see Supplementary information).

**Polyphenol Encapsulation within Coacervates.** The nano-coacervates were re-dispersed in 200 μL of bicine buffer at pH 8.5. Subsequently, the desired amount of TA in MQ water was added to the nano-coacervates, and the mixture was gently shaken (400 rpm) at 37 °C for 6 h. During this process, the color of the solvent was changed from white to yellowish due to TA oxidization. The resulting product was once again centrifuged at $3 \times g$ for 10 min to remove any unreacted TA molecules. The 0.03, 0.17, 0.33, and 1 mM of TA were incubated with nano-coacervates containing 1 mM of C5 peptide. Note that excess amounts of TA could lead to the formation of solid precipitates.

The micro-coacervates were first prepared by mixing 300 μL of heparin (100 U/ml) with 300 μL peptide (1.3 mg/ml) followed by the addition of TA molecules with bicine buffer overnight. The resulting product was centrifuged at $3 \times g$ for 10 min to remove any unreacted TA molecules. The pellet was re-dispersed in MQ water for future use. Note that excessive TA molecules can trigger solid aggregates (Supplementary Fig. 25).

**Sulfo-Cy5.5–Labeled C5 Peptides (i.e., C7).** Sulfo-Cy5.5-NHS was coupled with the free amine from the N-terminus of the C5 peptide to encapsulate sulfo-Cy5.5 dye within the nano-coacervates (see Supplementary information 2.2). Briefly, the desired amount of the C5 peptide was dissolved in DMSO with 1% v/v triethylamine wrapped with aluminum foil. Subsequently, sulfo-Cy5.5-NHS was added to the C5 peptides at a 1:1 molar ratio under generous stirring for 3 h. After the amine-NHS couplings, the final product was fully dried using a vacufuge. The resulting product was re-dispersed in 50% ACN for HPLC purification. MALDI-MOF MS was used to confirm the molecular weight of the final product (Supplementary Fig. 13).

For the encapsulation of C7, the nano-coacervates were dispersed in bicine buffer at pH 8.5, followed by the addition of C7 peptides under generous for 3 h. The resulting product was centrifuged at $3 \times g$ for 10 min to remove any unencapsulated C7 peptides. The pellet was re-dispersed in MQ water for future use.

**Thrombin Proteolysis of Nano-Coacervates.** Briefly, 1 mM of the nano-coacervates was incubated with various concentrations of α-thrombin ranging from 0.03 to 4 μM in 20 mM Tris-HCl buffer solution (pH 7.4, NaCl 150 mM). The mixture was immediately transferred into a 96-well plate, and the light absorption at 500 nm was measured at 37 °C for 1 h. The fragment solution was desalted by using a C18 column (5 μm, 9.4 × 250 mm) and then applied MALDI-TOF to confirm the cleavage site. Mass peaks shown in Fig. 2d are from the acetylated C5 parent and its fragment peptides.

Likewise, the C7-encapsulated nano-coacervates and NC-TA$_{0.13}$ were incubated in 50% human plasma containing the alpha thrombin (final concentration = 500 nM) at 37 °C. The mixture was immediately transferred into a 96-well plate, and the fluorescence signal at 670 nm was measured at 37 °C for 3 h.

**Stability test of NC-TAs.** Briefly, the nano-coacervates encapsulated with TA molecules (0.03–1 mM) were incubated with 1 M NaCl for 1 h at room temperature in a 96-well plate. The absorbance from 300 to

900 nm was measured with a step size of 2 nm before and after the incubation. The absorbances at 500 nm before and after incubation were used to calculate turbidity changes (Turbidity_after/Turbidity_before).

For the size measurements, NC-TA$_{0.13}$ was incubated for 1 h under various conditions including fibrinogen, glucose, glutamine, acetone, methanol (MeOH), citric acid (pH 2), DPBS, NaOH (pH 10), 60 °C, human albumin, NaCl of 150 mM, 50% of human serum, human urine, human saliva, and Dulbecco's modified Eagle's medium (DMEM), respectively. After incubation, the resulting samples were centrifuged at $3 \times g$ to replace the medium with MQ water for DLS measurement. The average size was calibrated using three independent replicates. The human saliva, serum, and urine samples were purchased from Innovative Research.

**C7 peptide/Heparin-FITC encapsulation and fluorescent monitoring in biofluids.** Briefly, NC-TAs were gently mixed with either C7 peptide (20 μM) or heparin-FITC (150 μM) for the encapsulation. The desired dye-conjugates were encapsulated within NC-TA$_0$ (i.e., nano-coacervates), NC-TA$_{0.07}$, and NC-TA$_{0.13}$, respectively with the same number of coacervates under generous stirring overnight. Note that the coacervate concentrations used for Figs. 3f and g are different. Following encapsulation, each sample was purified using centrifugation at $3 \times g$ for 10 min. Subsequently, the samples were transferred to a 96-well plate at 37 °C for PL measurement.

For fluorescence monitoring, NC-TAs encapsulating with either C7 peptide (Fig. 3f) or heparin-FITC (Fig. 3g) were incubated in 50% of human plasma in a final volume of 100 μL. The mixture was directly transferred to a 96-well plate at 37 °C, and the PL intensity at 670 nm (for C7 peptide) and 520 nm (for heparin-FITC) was recorded for 1 h with 1 min intervals. The activated PL signal indicates the disassembly of nano-coacervates in 50% human plasma. Notably, there was no self-PL quenching of C7 peptide or heparin-FITC in biofluids (Supplementary Fig. 33). The slope is referred to as the PL activation rate. Normal pooled plasma and normal pooled serum were purchased from Innovated Research.

**Cytotoxicity test and ROS detection of HUVEC.** HUVEC cells (ATCC, PCS-100-100) were cultured in a vascular cell basal medium with an endothelial cell growth kit. Cell cultures were incubated under 5% CO$_2$ at 37 °C. Cells were passaged when they reached 75–80% confluency using 0.25% trypsin for primary cells. DPBS and cell lysis buffer were used for negative and positive controls of healthy and dead cells.

For the cytotoxicity experiments, HUVEC cells were seeded overnight in a 96-well plate at a concentration of 10,000 cells/well. Subsequently, PBS, lysis buffer, C5 peptide, TA, nano-coacervates, NC-TA$_{0.25}$, NC-TA$_{0.5}$, and NC-TA$_1$ were co-incubated with HUVEC at an equal concentration of 0.16 mM for 12 h, respectively. A resazurin assay was used to analyze the cytotoxicity of nano-coacervates, NC-TA$_{0.25}$, NC-TA$_{0.5}$, and NC-TA$_1$ following a protocol. After 4 h incubation with resazurin, cell viability was calibrated by measuring the subtracted background absorbance of each well at 600 nm from resazurin absorbance at 570 nm. The absorbances of experimental wells were compared to those of the controlled wells containing healthy and dead cells.

HUVEC cells were seeded overnight in a 96-well plate for reactive oxygen species (ROS) detection test using a DCF-DA kit. Subsequently, PBS, lysis buffer, C5 peptide, TA, nano-coacervates, NC-TA$_{0.25}$, NC-TA$_{0.5}$, and NC-TA$_1$ were co-incubated with HUVEC at an equal concentration of 0.16 mM for 3 h, respectively. N-acetyl Cysteine and pyocyanin were added as the negative and positive controls. The fluorescence of experimental wells was compared to those of the controlled wells containing negative and positive controls. All experiments were performed in triplicate to measure the average and standard deviations.

**Reporting summary**

Further information on research design is available in the Nature Portfolio Reporting Summary linked to this article.

## Data availability

The data generated in this study are provided in the Supplementary Information/Source Data file. The data supporting the findings of this study are also available from the corresponding author upon request. Source data are provided with this paper.

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

## Acknowledgements

This work was supported by the National Institute of Health (DP2 HL137187 and R21 GM153048). W.Y. acknowledges the Schmidt Science Fellowship and the Diversity Fellowship from the UCSD Materials Science and Engineering program. W.Y. and C.B. thank the GEAR mentorship program at UC San Diego. This work was partly performed at the San Diego Nanotechnology Infrastructure (SDNI) of UCSD, supported by the National Science Foundation (Grant ECCS-2025752, and #2242375). The author acknowledges the use of Biorender software for scientific schematics (Fig. 1a).

## Author contributions

W.Y. and J.V.J. conceived the idea and developed the materials. W.Y. designed and performed major peptide synthesis and experimental works. Z.J. performed confocal imaging and ELISA. Z.J., Y.-C.C., M.N.C., T.H., Y.L., W.F.P., and J.Z. helped with other experimental work, sample collection, and scientific discussions. Y.-C.C. C.B. helped with peptide synthesis and purification. C.L., T.H., and M.R. helped with material characterizations. W.Y. and J.V.J. drafted the manuscript with input from all authors.

## Competing interests

The authors declare no competing interests
