## [Peer Review File · Nature Communications]

Polyphenols-Stabilized Coacervates for Enzyme-Triggered Drug DeliveryEditorial Note: Parts of this Peer Review File have been redacted as indicated to remove third-party material where no permission to publish could be obtained.

REVIEWER COMMENTS

Reviewer #1 (Remarks to the Author):

In this manuscript, the authors developed nano-sized coacervates via self-assembly of heparin glycosaminoglycan with mussel-inspired peptides (which contains thrombin-recognition site). Polyphenols were incorporated within the nano-coacervates and it maintains both stability and permeability. The designing droplets based on thrombin cleavage sites that can be externally modulated, and the disassembly rate of NC-TAs increased 4.2-fold with higher thrombin concentrations in human plasma, thus preventing the clotting cascade.

The paper is very well written. Tannic acid is a good membrane coating material, the polyphenolic membrane can be formed by spontaneous cross-linking (iron ions or alkaline conditions), and polyphenolic membrane encapsulation increases the stability of droplets. High concentrations of TA may lead to denaturation and inactivation of the enzyme. In this manuscript the authors have done a good job of determining the appropriate concentration of TA. This polyphenol- based approach not only improves stability but also opens the way for applications in biomedicine, protease sensing, and bio- responsive drug delivery.

A major revision is required and the following questions should be addressed.

1. As shown in Table 1, in order to simulate the natural products of mussels, the designed peptides contain two tyrosine residues. Why use two residues? What is their role in phase separation? Please explain whether the tyrosine residues participate the polymerization process as similar to the article (Peptide-Based Coacervate-Core Vesicles with Semipermeable Membranes; Adv. Mater. 2022, 34, 2202913).
2. It is better to investigate their fluidity of the coacervate droplets. The coacervates in this paper is nano-sized. The coacervate droplets are tend to calescence. Can the authors make a larger droplet (such as 5 microns) and test whether poly-tannic acid can improve its stability?
3. TA can self-aggregate at pH 8.5, and free TA can form aggregate nanoparticles. Please investigate if the droplets are the TA aggregates.
4. In a paper (Peptide-Based Coacervate Protocells with Cytoprotective Metal-Phenolic Network Membranes. J. Am. Chem. Soc. 2023, 145, 24108-24115), poly-TA as membranes were coated on the surface of coacervate. However, in this manuscript, TA is mainly distributed in the interior of the droplet. Are there any difference in the fabrication and chemical components, as compared with the JACS paper?

Reviewer #2 (Remarks to the Author):

General

The study delineates the properties of nano- to micro-sized coacervates comprised of heparin and

peptides containing tyrosine-lysine motifs. The incorporation of thrombin-cleavable peptide sequences between these motifs makes the particles enzyme-responsive, thereby allowing the adaptive release of the anticoagulant heparin according to the coagulation state. To prevent spontaneous disassembly in physiological media, the particles are stabilized with tannic acid (TA), a polyphenol crosslinking agent.

The authors meticulously analyze the effects of varying the number of tyrosine-lysine motifs in the peptide, and the concentrations of peptide and heparin on coacervate formation and thrombin responsiveness. The formation of a supramolecular network by TA is convincingly demonstrated. The particles' stability and responsiveness in biofluids, cytotoxicity assays, and anticoagulation tests in whole blood are thoroughly examined. However, the analysis of TA-stabilized coacervates lacks depth compared to that of the native ones, and some essential controls are missing.

The system developed by the authors represents an advancement in feedback-responsive anticoagulant materials, a topic discussed in references 28-32. Unlike other nanoencapsulated drug release systems, this system doesn't cause an immediate, complete release of the cargo upon degradation; instead, it tunes the release of heparin with the enzyme concentration. The use of tannic acid to stabilize the coacervates is a novel approach with potential for other applications.

Detailed

While plasma typically has no thrombin activity to prevent immediate clot formation, serum contains high residual thrombin activity due to the activation of the coagulation cascade. This necessitates an explanation of the results presented in lines 272-274. A thrombin quantification of the biofluids using chromogenic substrates is advisable.

The stability tests of the NC-TAs depend on the detection of the labeled peptide. However, the release profile of the peptide from the TA network may differ from that of heparin. Thus, quantifying the released heparin would provide more relevant information.

The whole blood assay should include a non-responsive control (Peptide C6), and the free heparin concentrations should be presented post-incubation. The study lacks an analysis of the particles' inflammatory potential and their uptake by granulocytes in whole blood. How long would the particles persist in whole blood or in circulation?

Minor

In line 109ff, the effects of varying heparin concentrations should be stated in the text, not just in the supplementary figure.

In line 65, the authors need to clarify how the same system can mitigate both the risk of thrombosis and excessive bleeding.

In line 88, the term "anti-thrombin" should be written as "antithrombin" to avoid confusion with an antibody.

Reviewer #3 (Remarks to the Author):

This study by Yim and Jokerst describes the development of heparin-peptide coacervates designed to achieve thrombin-triggered release of the anti-coagulant cargo. Coacervates were formed via a combination of electrostatic complexation and hydrogen bonding between heparin and tyrosine-arginine rich peptides. Later, a thrombin-recognition sequence was added to the peptide to enable enzyme-triggered release. Tannic acid was additionally required to stabilize the coacervates in physiologic solutions. The central premise is that, upon thrombin cleavage of the peptide, the coacervate disassociates to release heparin and effect anti-coagulant function. Overall, the manuscript is well written, and the authors employed an array of characterization methods to study the coacervates. However, the central innovation of the work, which is that the designed coacervates do not require a membrane for stabilization, is not sufficiently demonstrated. In addition, there are several instances where multiple techniques are employed without a clear research question to be answered or new information provided. Finally, the translational potential of the system was not sufficiently articulated or validated in a physiologically relevant clotting model. As a result, the work is too premature to warrant publication in Nature Communications.

Major issues/comments

1. The bio-inspired design of the coacervate peptide from mussels is overstated. Tyrosine in Mefp-5 is converted to DOPA via tyrosine hydroxylase, which is the functional amino acid that provides the adhesive properties of mussel foot pads. DOPA was not employed in the reported coacervate design.
2. It is not clear what additional insights are provided from the UV-Vis spectrum of Fig. 1c. Size determination was already presented in Fig. 1b, and the turbidity of the sample (which generates the broad absorption spectrum) is demonstrated in Figs. 1g and e.
3. Does the slight red-shift of Cy5 in Fig. 2e provide any biophysical context for the environment of the peptide in the coacervate? More discussion on this feature is needed to better understand how the peptide is incorporated within the coacervate matrix.
4. Controls are missing from the anti-coagulant studies presented. For example, free peptide is missing from assays presented in Fig. 2j and Fig. 4g. Similarly, free TA is required as a control in studies presented in Fig. 4g. It is possible that the non-heparin constituents of the coacervate can elicit anti-coagulant activity.
5. None of the results provided support a mechanism of TA encapsulation/incorporation within the

coacervate bulk, and could be readily explained by coating of the coacervate surface with the polyphenol. For example, addition of a TA surface coating to the coacervates would marginally increase their hydrodynamic size (expected to be ~1nm or less). Consequently, DLS measurements are not sufficient to deconvolute encapsulation vs. surface interactions. Similarly, UV-Vis (Fig. 3c), FTIR (Fig. 3d), TEM-HAADF and EDX (Fig. e – g), and tomography (Fig. 3i) do not provide any additional clarification on encapsulation vs. surface interactions. In fact, it is unclear the purpose of TEM-HAADF and EDX given that the peptide, heparin and TA have similar elemental constituents, and therefore no reasonable way to deconvolute their spatial distribution in the coacervate. This is an important flaw in the studies given that the central innovation of the work was to avoid requiring a surface membrane/coating to stabilize the coacervate.

6. A clear translational vision of the technology was not presented. Is the design intended to enable clot-specific release of encapsulated heparin? If this is the intended clinical context, rapid heparin release preferentially at the embolism site is desired. However, heparin release was found to be significantly hindered by incorporation of the TA (which in turn is needed for stability), and required ≥ 1 hour for complete heparin release. This raises significant concerns about the actual utility of the platform.

7. Related to the above point, experiments employing an in vitro/ex vivo thrombosis model that recapitulates the various physiologic factors of clotting (e.g., flow, presence of complement and clotting factors, etc.) is needed demonstrate the pre-clinical utility of the platform.

8. The purpose of testing coacervate stability in non-physiologically relevant acetone, methanol and concentrated citric acid solvents is not articulated.

9. HEK293 cells are not a suitable model to study off-target endothelial toxicity. An actual endothelial cell line (e.g., HUVEC) should be employed to look at potential cardiovascular toxicity.

To help the legibility of the remainder of this response letter, all reviewer's comments and questions are typeset in black bold font. Changes made in our revised manuscript are written in blue bold font. All changes to the original document are highlighted. We also include a clean version. We hope you will find our revised manuscript suitable for publication in Nature Communications.

Reviewer #1 (Remarks to the Author):

In this manuscript, the authors developed nano-sized coacervates via self-assembly of heparin glycosaminoglycan with mussel-inspired peptides (which contains thrombin-recognition site). Polyphenols were incorporated within the nano-coacervates and it maintains both stability and permeability. The designing droplets based on thrombin cleavage sites that can be externally modulated, and the disassembly rate of NC-TAs increased 4.2-fold with higher thrombin concentrations in human plasma, thus preventing the clotting cascade.

The paper is very well written. Tannic acid is a good membrane coating material, the polyphenolic membrane can be formed by spontaneous cross-linking (iron ions or alkaline conditions), and polyphenolic membrane encapsulation increases the stability of droplets. High concentrations of TA may lead to denaturation and inactivation of the enzyme. In this manuscript, the authors have done a good job of determining the appropriate concentration of TA. This polyphenol-based approach not only improves stability but also opens the way for applications in biomedicine, protease sensing, and bio-responsive drug delivery.

We thank the reviewer for the highly positive compliment and the enthusiasm for the work.

1. As shown in Table 1, in order to simulate the natural products of mussels, the designed peptides contain two tyrosine residues. Why use two residues?

Thank you for the point. We designed our peptide building block that can be disassembled by thrombin cleavage. One YR unit is not sufficient to trigger coacervation. In contrast, four repeating YR units would make a fragment that still has at least two YR units capable of forming coacervates as depicted in Table 1 and Fig. 1d. We have included the rationale for using two repeating YR residues on Page 8.

We envisioned that if thrombin can cleave the peptide building block and disassemble the coacervates, then the coacervates could be an enzyme-responsive platform that can release the encapsulated heparin in response to thrombin proteolysis (Fig. 2a). To achieve this, we added a thrombin cleavage site (LVPR↓GS)¹ in the middle of the C2 sequence: YRLVPRGSYR (referred to as C5) (Table 1). Thrombin proteolysis would thus result in fragment peptides that only contain one YR unit which is not sufficient for phase separation as depicted in Fig. 1d.

2. What is their role in phase separation? Please explain whether the tyrosine residues participate in the polymerization process as similar to the article (Peptide-Based Coacervate-Core Vesicles with Semipermeable Membranes; Adv. Mater. 2022, 34, 2202913).

The role of tyrosine (Y) in phase separation is to facilitate hydrophobic interactions between phenolic groups. The role of arginine (R) is to provide a positive charge for electrostatic interactions with heparin. The suggested article (**Peptide-Based Coacervate-Core Vesicles with Semipermeable Membranes; Adv. Mater. 2022, 34, 2202913**) uses tyrosine-rich peptide to induce phase separation by pH change, and it does not involve any cargo and has no charge domain for phase separation. In contrast, our coacervates are designed to deliver heparin glycosaminoglycan. We used tyrosine and arginine repeating units to form coacervates with heparin via electrostatic and hydrophobic interactions.

Note that heparin is known to interact with diverse proteins through electrostatic interactions and non-ionic strengths such as hydrogen bonding.² Small molecular dyes interact with heparin through π - π stacking, hydrophobic, and electrostatic interactions confirmed by computational and experimental methods.³

[REDACTED]

Panel C shows the π - π stacked MB dimer bound on the sulfate and glucosamine. The fraction of MB dimers formed by π - π stacking (distance between two MB molecules was less than 4.0

Å) among the total number of MB dimers (distance between two MB molecules was less than 9.5 Å) is shown in panel D. Figure and descriptions were from **Wang J, et al Bioconjugate Chemistry 2018, 29 (11), 3768–3775.**

Thus, we utilized the mussel-inspired YR-based peptide sequence to make heparin coacervation through electrostatic and hydrophobic interactions. We include this description on Page 9

Our previous studies revealed that heparin could assemble with small molecular dyes via strong electrostatic and hydrophobic interactions^{3,4} suggesting that the repetitive YR peptide sequence might also readily trigger the formation of coacervates. To test our hypothesis, we synthesized a short YRYR peptide (referred to as C2) and mixed C2 (0.05 to 1.5 mM) with heparin (50 U/ml). Upon

interaction with heparin, the C2 peptide instantly formed coacervate droplets of varying sizes confirmed by dynamic light scattering (DLS) (Fig. 1b and Fig. S2).

Importantly, we conducted a disassembly test in different conditions and pH to examine which interactions governed our coacervate system. Electrostatic, pi-pi interactions and hydrogen bonding are involved in our coacervate system. We have mentioned this information on Page 5.

Stability test of nano-coacervates in different pH (g) and different conditions (h) including PEG2000, citric acid, urea, triton, SDS, DMF, and DMSO.

To confirm the interactions governing the coacervate formation of C2–heparin complex, nano-coacervates were incubated with PEG2000, citric acid, urea, TritonX-100, SDS, DMF, and DMSO, respectively. TritonX-100 and SDS can break non-ionic or ionic interactions; DMSO and DMF are organic solvents that can destroy pi-pi interaction;⁵ and urea can break the hydrogen bonding.⁶ The nano-coacervates were disassembled in urea, TritonX-100, SDS, DMF, and DMSO conditions, indicating that multiple interactions including electrostatic, pi-pi interaction, and hydrogen bonding were involved in the formation of C2–heparin coacervates (Fig. 1h and Fig. S10). Nano-coacervates were stable at low pH (1–5), but they disassembled at high pH (over 9) due to the deprotonation of the guanidine group (Fig. 1i). The isoelectric point of arginine is 10.8.⁷

2. It is better to investigate the fluidity of the coacervate droplets. The coacervates in this paper is nano-sized. The coacervate droplets tend to calescence. Can the authors make a larger droplet (such as 5 microns) and test whether poly-tannic acid can improve its stability?

Good point. We performed polyphenol encapsulations in a large droplet and discovered that polyphenol-stabilized micro-coacervates could endure centrifugation at 3 x g and showed high stability over time. We now add this information in Fig S27 on Page 12.

We also discovered that polyphenol encapsulation can be applied to micro-sized coacervates (MC-TAs), and improved colloidal stability. The optical image illustrates the uniformly dispersed micro-coacervates after TA encapsulation (Fig. S25). The spherical shapes and sizes of the dried MC-TA_{0.05} were confirmed by the SEM technique (Fig. 3k and Fig. S26). Notably, we observed significantly improved colloidal stability under the centrifugation of 3 x g. The coacervate droplets without TA showed a 98.7% decrease in turbidity while MC-TA_{0.05} decreased by only 1.4% (Fig. S27).

Fig S27: TA-encapsulated of micro-coacervates (MC-TAs)

a) UV-vis spectra of MC-TAs and TA only at the concentrations of 0.05 and 0.13 mM. MC-TAs increased extinction ranging from 400 to 900 nm due to the formation of coacervate droplets while TA alone has minimal extinction. Decrease in the extinction of coacervate droplets without TA (b) and MC-TA_{0.13} (c) before and after 3xg centrifugation for 10 min. The coacervate droplets without TA showed a 98.7% decrease in turbidity while MC-TA_{0.05} decreased by only 1.4%. These results indicate that TA encapsulation improves the stability of not only nano-sized but also micro-sized coacervate droplets. d) DLS data of MC-TA_{0.13}. e) Turbidity of MC-TA_{0.13} at different time points, showing its high stability. The turbidity was calibrated based on the extinction value (the equation is described in Supporting information 2.14)

Panel a in Fig. S25 shows coacervate images before and after the addition of TA. Panels b and c used 0.04 mM and 0.16 mM of TA, respectively.

Fig. S25: Optical images of micro-coacervates after polyphenol encapsulation
 Optical images of coacervate droplets before (a) and after TA encapsulation of 0.04 mM (b) and 0.16 mM (c). The images were collected after the centrifugation. After centrifugation, the average diameter of MC-TA was $2.01 \pm 0.44 \mu\text{m}$. The average and standard deviation represent 16 independent droplets. The scale bar represents 20 μm .

We also used SEM techniques to image these micro-sized coacervates encapsulated polyphenols. We added all the information in Fig. 3 on Page 13.

Fig. S26: SEM images of micro-coacervates
 SEM images of low (a) and high (b) concentrations of MC-TAs with different magnifications. TA concentration with 0.16 mM was used for encapsulation. The image in the inset shows the spherical shapes of MC-TAs and size distributions.

The average diameters of MC-TAs were $2.01 \pm 0.44 \mu\text{m}$ when hydrated and $0.38 \pm 0.9 \mu\text{m}$ when evaporated, respectively. The average diameter was calibrated using 16 individual MC-TAs from optical and SEM images, respectively. The average and standard deviation represent 16 independent droplets.

Also, we describe the experimental methods for the TA encapsulation process within micro-coacervates on Page 18.

The micro-coacervates were first prepared by mixing 300 μL of heparin (100 U/ml) with 300 μL peptide (1.3 mg/ml) followed by the addition of TA molecules with bicine buffer overnight. The resulting product was centrifuged at 3 g for 10 min to remove any unreacted TA molecules. The pellet was re-dispersed in MQ water for future use. Note that excessive TA molecules can trigger solid aggregates (Fig. S25).

3. TA can self-aggregate at pH 8.5, and free TA can form aggregate nanoparticles. Please investigate if the droplets are the TA aggregates.

We agreed that TA can trigger particle aggregation. Indeed, it is important to use desired amounts of tannic acid for encapsulation; otherwise, it will aggregate coacervates. We observed that *excess polyphenols can aggregate the micro-sized coacervates* seen in panel d below. Therefore, we examined an optimal point between the concentrations of coacervates and the amounts of TA for encapsulations. After encapsulation, free TA was removed and pH was changed to pH 7 to prevent self-polymerization. We described all the information in the experimental section on Page 18.

d) A high concentration of micro-coacervates incubated with different amounts of TA from 0.04 to 0.2 mM overnight. The red-dotted area indicates increased contrast between coacervate droplets due to high TA concentrations. The addition of 0.2 mM TA resulted in the formation of solid aggregates. The images were obtained before centrifugations.

Therefore, we used appropriate amounts of TA (before we observed aggregates) and removed all free TA after encapsulation by centrifuge. Our DLS data profiled the narrow size distributions (PDI <0.01) of NC-TAs in Fig. 3 and maintained its proteolytic activity (Fig. 4). In contrast, DLS failed to profile TA aggregates only. We add all the discussion on Page 12.

Fig. 3: Polyphenols-encapsulated coacervates a) Schematic illustration of TA encapsulation within the coacervates. DLS (b), UV-vis spectra (c), and FTIR data (d) of NC-TAs. The brown regions in (c) and (d) indicate the appeared peaks after TA encapsulation. Stability test of NC-TAs in different pH (e) and different conditions (f) including PEG2000, citric acid, urea, triton, SDS, DMF, and DMSO. g) TEM image of NC-TA_{0.13}. h) TEM and HAADF images of a single NC-TA_{0.13} at different angles. Fig. S22–S24 show multiple NC-TA_{0.13} at different angles with low magnification. The scale bars represent 100 nm.

We have detailed all the information from Fig. 3 and now include them in the experimental methods section on Page 18.

The nano-coacervates were re-dispersed in 200 μL of bicine buffer at pH 8.5. Subsequently, the desired amount of TA in MQ water was added to the nano-coacervates, and the mixture was gently shaken (400 rpm) at 37C for 6 h. During this process, the color of the solvent was changed from white to yellowish due to TA oxidization. The resulting product was once again centrifuged at 3 x g for 10 min to remove any unreacted TA molecules. The 0.03, 0.17, 0.33, and 1 mM of TA were incubated with nano-coacervates containing 1 mM of C5 peptide. Note that excess amounts of TA could lead to the formation of solid precipitates.

4. In a paper (Peptide-Based Coacervate Protocells with Cytoprotective Metal-Phenolic Network Membranes. J. Am. Chem. Soc. 2023, 145, 24108-24115), poly-TA as membranes were coated on the surface of coacervate. However, in this manuscript, TA is mainly distributed in the interior of the droplet. Are there any differences in the fabrication and chemical components, as compared with the JACS paper?

The paper (Peptide-Based Coacervate Protocells with Cytoprotective Metal-Phenolic Network Membranes. J. Am. Chem. Soc. 2023, 145, 24108-24115) utilized Metal-Phenolic Network (MPN) to form a membrane on the surface of coacervates. Polyphenols were encapsulated within the coacervates followed by the addition of metal ions, forming the MPN membrane. However, our fabrication system *does not include any metal ions for the MPN membrane*. Our coacervate maintains permeability for thrombin. Furthermore, the paper used R10/D10 peptides to form complex coacervates while our work involves *alternating arginine and tyrosine sequences* to form coacervates with heparin glycosaminoglycan.

TEM and EDX Images just below are sourced from (J. Am. Chem. Soc. 2023, 145, 24108-24115), showing a MPN membrane only after removing the interior using a DMSO.

[REDACTED]

TEM and EDX images of NC-TAs below are sourced from Fig. 4i, and they show uniformly distributed elemental signals from the inside. We also confirmed that DMSO *completely removed our coacervates*.

(i-j) EDX elemental mapping of a single NC-TA_{0.13}, showing C, N, O, and S elements which are components of heparin, peptide, and TA.

To investigate whether TA is distributed in the interior of the droplet, we conjugated coumarin boronic acid with TA and then encapsulated TA-coumarin conjugates inside micro-sized coacervates for confocal imaging. The confocal image (Fig. 3I) shows the uniformly distributed fluorescent signal arising from inside coacervates, indicating TA encapsulation rather than membrane coating. We now describe all the information on Page 12.

Lastly, coumarin boronic acid was conjugated with TA for confocal imaging to verify TA encapsulation within coacervate droplets. HPLC was utilized to remove free coumarin dyes from TA-coumarin before encapsulation (Fig. S28). Fig. 3I shows a uniformly distributed fluorescent signal of TA-coumarin conjugates from inside the MC-TAs. This result indicates that TA is encapsulated within the coacervates rather than being membrane-coated.^{8,9}

I) confocal image of micro-coacervates encapsulating TA-coumarin conjugates. The yellow box highlights the evenly distributed fluorescent signal of TA-coumarin inside a single MC-TA. Coumarin boronic acid was linked with hydroxyl groups in TA, forming a boronate ester. The conjugates were purified using HPLC before encapsulation.

[REDACTED]

The confocal image on the right is our micro-coacervates encapsulated TA (MC-TAs), showing a uniformly distributed fluorescent signal. In contrast, the image on the left (from *ACS Nano* 2023, 17, 17, 16980–16992) is coacervates coated with DPPS-Au/Ag clusters on the membrane, showing a ring.

These new findings are now available in Fig. 3 on Pages 12-13, and we cite two relevant papers that show membrane coating (showing a ring) on the coacervates for comparison.

Reviewer #2 (Remarks to the Author):

General

The study delineates the properties of nano- to micro-sized coacervates comprised of heparin and peptides containing tyrosine-lysine motifs. The incorporation of thrombin-cleavable peptide sequences between these motifs makes the particles enzyme-responsive, thereby allowing the adaptive release of the anticoagulant heparin according to the coagulation state. To prevent spontaneous disassembly in physiological media, the particles are stabilized with tannic acid (TA), a polyphenol crosslinking agent. The authors meticulously analyze the effects of varying the number of tyrosine-lysine motifs in the peptide, and the concentrations of peptide and heparin on coacervate formation and thrombin responsiveness. The formation of a supramolecular network by TA is convincingly demonstrated. The particles' stability and responsiveness in biofluids, cytotoxicity assays, and anticoagulation tests in whole blood are thoroughly examined.

1. However, the analysis of TA-stabilized coacervates lacks depth compared to that of the native ones, and some essential controls are missing.

We now include three major discoveries of TA-stabilized coacervates. First, NC-TAs showed higher stability in pH 10 and other solvent conditions including urea compared to nano-coacervates only. Second, the polyphenol strategy can be applied to *micro-sized coacervates*, also enhancing stability. Third, we conjugated polyphenols (i.e., TA) with coumarin boronic acid and then encapsulated the TA-coumarin conjugates within micro-coacervates for confocal imaging. We observed uniformly distributed strong fluorescent signals from inside coacervates. Confocal imaging shows that *polyphenols were encapsulated inside the nano-coacervates rather than being membrane-coated*.

[REDACTED]

The confocal image on the right is our coacervates encapsulating TA-coumarin conjugates that show a uniformly distributed fluorescent signal. In contrast, the image on the left (from *ACS Nano* 2023, 17, 17, 16980–16992) is coacervates coated with DPPS-Au/Ag clusters on the membrane, showing a ring.

All these new findings are now available in Fig. 3 (see below) on Pages 12-13, and we cite two relevant papers that show membrane coating on the coacervates for comparison.

Fig. 3: Polyphenols-encapsulated coacervates a) Schematic illustration of TA encapsulation within the coacervates. DLS (b), UV-vis spectra (c), and FTIR data (d) of NC-TAs. The brown region in (c) and (d) indicates the appeared peaks after TA encapsulation. Stability test of NC-TAs in different pH (e) and different conditions (f) including PEG2000, citric acid, urea, triton, SDS, DMF, and DMSO. g) TEM image of NC-TA_{0.13}. h) TEM and HAADF images of a single NC-TA_{0.13} at different angles. Fig. S22–S24 show multiple NC-TA_{0.13} at different angles with low magnification. The scale bars represent 100 nm. (i-j) EDX elemental mapping of a single NC-TA_{0.13}, showing C, N, O, and S elements which are components of heparin, peptide, and TA. The red-dotted line indicates the region used for EDX mapping. k) SEM of MC-TAs. l) confocal image of micro-coacervates encapsulating TA-coumarin conjugates. The yellow box highlights the evenly distributed fluorescent signal of TA-coumarin inside a single MC-TA. Coumarin boronic acid was linked with hydroxyl groups in TA, forming a boronate ester. The conjugates were purified using HPLC before encapsulation. The experiments were performed in triplicate.

The system developed by the authors represents an advancement in feedback-responsive anticoagulant materials, a topic discussed in references 28-32. Unlike other nanoencapsulated drug release systems, this system doesn't cause an immediate, complete release of the cargo upon degradation; instead, it tunes the release of heparin with the enzyme concentration. The use of tannic acid to stabilize the coacervates is a novel approach with potential for other applications.

We appreciate the reviewer for the positive compliments and good suggestions.

Detailed

2. While plasma typically has no thrombin activity to prevent immediate clot formation, serum contains high residual thrombin activity due to the activation of the coagulation cascade. This necessitates an explanation of the results presented in lines 272-274. A thrombin quantification of the biofluids using chromogenic substrates is advisable.

Thank you for your points. We now utilize thrombin chromogenic substrates to measure residual thrombin activity in human plasma and serum, respectively. We agree that human plasma tends to have less residual thrombin activity compared to serum samples because human plasma is obtained using anticoagulants which prevent clotting cascade and thus inhibit thrombin formation. In contrast, human serum is obtained by allowing blood to clot, during which thrombin is generated from prothrombin. We observed that human serum exhibited 20-fold higher optical density from residual thrombin activity compared to human plasma (see Fig. 4j below on the left).

Lastly, we examined the residual thrombin activity of human serum and plasma using a thrombin chromogenic substrate. Human serum exhibited a 20-fold higher residual thrombin activity than plasma (Fig. 4j). The residual thrombin activity in human serum was comparable to that of 42.5 nM of the alpha-thrombin. This difference arises because human plasma is obtained by anticoagulating blood, which prevents the clotting cascade, thus inhibiting thrombin formation. In contrast, human serum is collected by allowing blood to clot, during which thrombin is generated from prothrombin, resulting in higher residual thrombin activity. Nano-coacervates without TA showed a 56% decrease in absorbance at 500 nm due to residual thrombin activity while NC-TA_{0.13} showed only an 18% decrease, indicating enhanced stability through polyphenol encapsulations (Fig. S37).

j) Residual thrombin activity in human serum and plasma. Human serum shows higher residual thrombin activity comparable to 42.5 nM of alpha-thrombin (Fig. S37). The graphs on the right in panel (j) represent absorbance changes of nano-coacervates and NC-TA_{0.13} before and after 1h incubation in human serum, showing higher stability of NC-TA_{0.13} than pristine nano-coacervates.

In addition, we also used a thrombin chromogenic substrate to measure a standard curve of *alpha-thrombin* shown in Panel A in Fig. S37 (see below). The obtained fitting curve was used to convert the optical density of human serum samples into corresponding thrombin concentrations. The data suggests that human serum exhibited similar thrombin activity to that of 42.5 nM alpha-thrombin. This residual thrombin activity may be involved in the disassembly of nano-coacervates in human serum while NC-TAs still maintained higher stability. NC-TAs showed a reduced decrease in absorbance by 1.86-fold than pristine nano-coacervates. We now have included all this information in Fig. 4j on Page 15.

Fig. S37: Residual thrombin activity and stability test of NC-TA_{0.13} in human serum
a) A standard curve indicates residual thrombin activity of alpha thrombin in different concentrations measured by thrombin chromogenic substrate. **b)** Optical density (i.e., residual thrombin activity) of human serum samples with serial dilution. Different dilution factors (1:2, 1:4, 1:8, 1:16, and 1:32) were examined, showing that human serum contained residual thrombin activity comparable to that of thrombin at a concentration of 42.5 nM. UV-vis spectra of pristine nano-coacervates **(c)** and NC-TA_{0.13} **(d)** in 50% human serum after 1h incubation.

3. The stability tests of the NC-TAs depend on the detection of the labeled peptide. However, the release profile of the peptide from the TA network may differ from that of heparin. Thus, quantifying the released heparin would provide more relevant information.

Good point. We now include tracking heparin together with C7 peptides. To understand heparin release, we further made fluorescent-tagged heparin with FITC (material information is shown on the next page). We used this fluorescent analogue to make coacervates and then monitored heparin release after thrombin activation via fluorescein signal. The addition of thrombin promptly increased the disassembly rate (i.e., PL activation). PL intensity of heparin-FITC turned on within 10 min as shown in Fig. 4g. We now include all the information on Page 15. We also include the experimental details on Page 20: experimental section 1.6.

Both pristine nano-coacervates and NC-TA_{0.13} containing C7 peptides and heparin-FITC were incubated in 50% human plasma, respectively. The quenched fluorescence of the C7 peptide and heparin-FITC was activated as a function of the disassembly of nano-coacervate (Fig. 4e). NC-TA_{0.13} exhibited a 3.3-fold decrease in the PL activation rate of C7 peptide than NC-TA₀, indicating enhanced stability in human plasma. Simultaneously, thrombin could accelerate heparin release from NC-TA_{0.13} (Fig. 4f). The disassembly rate of NC-TA_{0.13} increased by 4.2-fold upon the addition of thrombin (500 nM) which falls within the physiologic range of free thrombin concentration. Physiologic concentrations of free thrombin during coagulation reactions range over 500 nM.¹⁰ We also monitored this disassembly process using heparin-FITC (Fig. S32). NC-TAs exhibited a decrease in PL activation of heparin-FITC compared to pristine nano-coacervates in human plasma. Concurrently, the addition of thrombin rapidly increased the PL activation rate of heparin-FITC by 1.8-fold, recovering PL intensity within 10 min (Fig. 4g). There was no fluorescence quenching of C7 peptides and heparin-FITC either by the background medium (*i.e.*, human plasma) or by TA molecules (Fig. S33).

e) Schematic illustration of monitoring C7 peptide (f) and heparin-FITC (g) during disassembly of NC-TAs by thrombin. The left panels in (f-g) show a decrease in the PL activation rate of NC-TAs compared to nano-coacervates (*i.e.*, NC-TA₀) due to improved stability in 50% human plasma. The right panels in (f-g) illustrate the

addition of thrombin rapidly activates the PL intensity of C7 peptide or heparin-FITC, indicating proteolysis-driven heparin release.

Here is the material information about Heparin-FITC. Heparin-FITC exhibited Ex/Em wavelength in 494/518 nm as described in Fig. S32.

Fig. S32. Heparin Fluorescein (FITC)

Molecular structure of heparin-FITC, Mw 27k (Creative PEGWorks, NC, USA), and its absorbance (b) and fluorescence (c). Heparin-FITC encapsulated within the NC-TAs was monitored in 50% human plasma with and without thrombin described in Fig. 4g. Thrombin accelerated the disassembly of NC-TAs, rapidly turning on the C7 peptide and heparin-FITC, respectively.

4. The whole blood assay should include a non-responsive control (Peptide C6), and the free heparin concentrations should be presented post-incubation.

Good point. We further tested non-responsive control (scramble peptides), peptide, tannic acid, and free heparin with different concentrations that were post-incubated. We collected fresh blood in EDTA-treated blood collection tubes. Then, we added calcium chloride to trigger blood coagulation for the test. First, different heparin concentrations were used to check anticoagulation activity as described in Fig. S35.

Fig. S35: Standard curve and F1+2 peptide concentration in human plasma
a) A standard curve of prothrombin fragment F1+2 concentration. Activated optical density (O.D.) at 450 nm was converted to F1+2 concentrations using an equation from the fitting curve. b) F1+2 fragment concentrations from whole human blood at different heparin concentrations from 0.2 to 1 U/ml., showing anticoagulation activity of free heparin. Whole human blood was collected using an EDTA-treated blood collection tube. Calcium chloride was used to trigger blood coagulation. 0.4 mL of whole human blood was incubated with different heparin concentrations. Low heparin concentration (lower than 0.1 U/ml) induced the strong thrombus, increasing F1+2 fragment concentrations due to thrombin activation.

After confirming the anticoagulation activity of free heparin, we then created NC-TA_{0.13} made of C6 (scramble) and C5 (thrombin-active) peptides, respectively. Then, we incubated heparin only, NC-TA_{0.13}, scramble NC-TA_{0.13}, tannic acid, and peptide only with whole human blood at the same concentrations (0.6 U/ml) for 15 min to examine blood coagulation. *Scramble NC-TA_{0.13}, TA, and C6 peptide only showed strong thrombus while heparin and NC-TA_{0.13} prevented blood coagulation as shown in Fig. 4i.* We now include all non-responsive controls on Page 15.

We tested the anticoagulant performance of NC-TA_{0.13} in whole human blood using a human prothrombin fragment 1+2 (F1+2) enzyme-linked immunosorbent assay (ELISA) kit. Fresh whole blood was collected using an EDTA-treated blood collection tube. Free heparin (0.6 U/ml), C6 peptide, TA, NC-TA_{0.13}, and scramble NC-TA_{0.13} were incubated with whole human blood at the same concentration, respectively. Calcium chloride was then used to trigger blood clot formation. We observed a negligible difference in F1+2 formation between heparin and NC-TA_{0.13}, confirming active blood anticoagulation driven by heparin released from NC-TA_{0.13} (Fig. 4i and Fig. S35-S36). In contrast, a strong thrombus was observed in TA, C5 peptide only, and scramble NC-TA_{0.13}. Note that scramble NC-TA_{0.13} was comprised of a scramble C6 peptide, serving as a non-responsive control.

i) Prothrombin F1+2 peptide concentrations of NC-TA_{0.13} TA, C5 peptide, and NC-TA_{0.13} made of scramble peptide (*i.e.*, C6) from whole human blood incubation. The inserted photo shows a strong blood clot from scramble NC-TA_{0.13} (right) and blood anticoagulation from NC-TA_{0.13} (left).

5. The study lacks an analysis of the particles' inflammatory potential and their uptake by granulocytes in whole blood.

Thank you for your comments. We now test the particles' inflammation potential and cell viability using DCF-DA and viability assays. A new cell line (HUVEC) was further used to examine the potential for toxicity. HUVECs exhibited high cell viability and minimal ROS intensity when exposed to NC-TAs at 0.16 mM concentration. We also tested its structural components including TA, heparin, and C5 peptide. Future work will investigate cell uptake in animal models. All the information is now available in Fig. 4h and described on Page 15. Detailed experimental methods are described on Page 20.

We next examined the cell viability and cellular reactive oxygen species (ROS) levels of NC-TA_{0.13} using human umbilical vein endothelial cells (HUVECs), respectively. An endothelial cell line (e.g., HUVEC) was utilized to examine the potential for cardiovascular toxicity. NC-TAs and their structural components including C5 peptide, heparin, and TA exhibited high cell viability (>83%) and minimal ROS intensities (Fig. 4h).

h) Cell viability and ROS intensity of HUVEC incubating with PBS, TA, heparin, C5 peptide, and NC-TAs, respectively.

6. How long would the particles persist in whole blood or circulation?

We coated NC-TA_{0.13} on inert material, and the coated NC-TA_{0.13} showed exceptional stability in whole human blood (100%) for 1h as shown in Fig. S38. Notably, our peptide building block and tannic acid contained phenol chemical moieties, enabling strong surface coating on inert materials.

Fig. S38: Optical images of MC-TAs before and after blood incubation. TA-encapsulated micro-coacervates (MC-TAs) were coated on the glass slides. a) Optical images show the MC-TAs, maintaining spherical shapes on the glass slides after drying and incubating in PBS and DMEM for 1 h. These MC-TAs showed

exceptional stability after incubating in whole human blood (100%) (b). The red-dotted circle represents a single MC-TA before and after the incubation. The scale bar represents 50 μm .

The *coacervate coating* holds a strong potential in drug delivery systems not only recently shown in gastrointestinal diseases¹¹ We envision that our coacervates strengthened by TA can be integrated into a medical device for a novel drug delivery platform such drug-eluting stent for cardiovascular interventions. We demonstrate that our polyphenol-coacervates also exhibit exceptional coating ability on inert materials with high stability in whole human blood. We now add all the discussion on Page 17.

Coacervates containing catechol as structural building blocks have shown significant potential in drug delivery systems, particularly for gastrointestinal diseases,¹¹ due to their strong adhesiveness¹² capable of prolonged retention in the gastrointestinal tract. Our nano-coacervates strengthened by polyphenols also exhibit superior coating ability on inert substrates and maintain high stability in whole human blood (Fig. S38). Future work will involve incorporating coacervates on medical devices such as a drug-eluting stent to facilitate effective anticoagulant delivery for cardiovascular interventions.

Minor

In line 109ff, the effects of varying heparin concentrations should be stated in the text, not just in the supplementary figure.

We now include the effects of varying heparin concentrations in Fig. 1 and make this point clear in the text.

In addition, various heparin concentrations from 0.25 to 50 U/ml were combined with a constant peptide concentration of 1 mM, confirming that the coacervation relies on the number of YR units and heparin concentration (Fig. 1e and Fig. S8).

Formation of coacervates as a function of different peptide (d) and heparin (e) concentrations.

In line 65, the authors need to clarify how the same system can mitigate both the risk of thrombosis and excessive bleeding.

We now include information to clarify how incorporating the molecular feedback regulation system of hemostasis into the drug delivery system could mitigate the risks of thrombosis and excessive bleeding on Page 4.

Living organisms maintain hemostasis through precise molecular feedback regulations. For example, vascular injury triggers a coagulation cascade process where clotting factors activate prothrombin to thrombin, which transforms fibrinogen into insoluble fibrin by cleavage. Together with platelet activation, this process produces stable fibrin clots to prevent excessive bleeding.¹³ We envision that by incorporating a feedback loop system within the coacervates, they could

regulate heparin release based on thrombin activity. Increasing environmental thrombin levels would promptly trigger heparin release, while normal physiological thrombin levels would leave the coacervates intact—thus, there would be no risk of excessive bleeding.^{14,15}

In line 88, the term "anti-thrombin" should be written as "antithrombin" to avoid confusion with an antibody.

We correct the word and now written as antithrombin.

Reviewer #3 (Remarks to the Author):

This study by Yim and Jokerst describes the development of heparin-peptide coacervates designed to achieve thrombin-triggered release of the anti-coagulant cargo. Coacervates were formed via a combination of electrostatic complexation and hydrogen bonding between heparin and tyrosine-arginine rich peptides. Later, a thrombin-recognition sequence was added to the peptide to enable enzyme-triggered release. Tannic acid was additionally required to stabilize the coacervates in physiologic solutions. The central premise is that, upon thrombin cleavage of the peptide, the coacervate disassociates to release heparin and effect anti-coagulant function. Overall, the manuscript is well written, and the authors employed an array of characterization methods to study the coacervates. However, the central innovation of the work, which is that the designed coacervates do not require a membrane for stabilization, is not sufficiently demonstrated. In addition, there are several instances where multiple techniques are employed without a clear research question to be answered or new information provided. Finally, the translational potential of the system was not sufficiently articulated or validated in a physiologically relevant clotting model. As a result, the work is too premature to warrant publication in *Nature Communications*.

We thank the reviewers' comments and suggestions and have revised the manuscript accordingly including a physiologically relevant clotting model, which underscores the value of the activatable heparin system.

Major issues/comments

1. The bio-inspired design of the coacervate peptide from mussels is overstated. Tyrosine in Mefp-5 is converted to DOPA via tyrosine hydroxylase, which is the functional amino acid that provides the adhesive properties of mussel foot pads. DOPA was not employed in the reported coacervate design.

Thank you for your points. We now termed our peptide design as tyrosine and arginine-based peptide and corrected that Mefp-5 consists of DOPA and lysine. We also clarified the structural difference between DOPA and tyrosine: catechol and phenol groups in Fig. 1a.

The *Mytilus edulis* foot protein 5 (Mefp-5) in mussels contains repetitive DOPA and lysine (K) groups that provide positively charged residues with hydrophobic interactions.¹⁶ This enables Mefp-5 to interact with a wide array of materials through either covalent or noncovalent interactions (Fig. S1).¹⁷⁻¹⁹ The first step of designing our system was to determine whether a short peptide composed of tyrosine (Y) and arginine (R) could form a coacervate droplet with heparin (average M_w : 15,000 Da) (Fig. 1a).

a) Schematic illustration of coacervate design using heparin and a YR-based short peptide.

Our peptide design containing both tyrosine and arginine shares a similar concept with Mefp-5 structures containing DOPA and lysine (Fig. S1). The combination of electrostatic and hydrophobic interactions allows Mefp-5 to interact with diverse materials. Likewise, we used positively charged arginine and hydrophobic tyrosine to trigger coacervation with heparin. We also showed that our coacervate formations involve multiple interactions including electrostatic, pi-pi stacking, and hydrogen bonding in Fig. 1h. We have included all the information and clarified structural differences to avoid overstatement in Fig. 1 on Page 5.

Heparin is a glycosaminoglycan with repeating sulfate units that provide a negative charge and a polysaccharide structure for efficient binding with antithrombin.²⁰ Our previous studies revealed that heparin could assemble with small molecular dyes *via* strong electrostatic and hydrophobic interactions^{3,4} suggesting that the repetitive YR peptide sequence might also readily trigger the formation of coacervates. To test our hypothesis, we synthesized a short YR YR peptide (referred to as C2) and mixed C2 (0.05 to 1.5 mM) with heparin (50 U/ml). Upon interaction with heparin, the C2 peptide instantly formed coacervate droplets of varying sizes confirmed by dynamic light scattering (DLS) (Fig. 1b and Fig. S2).

2. It is not clear what additional insights are provided from the UV-Vis spectrum of Fig. 1c. Size determination was already presented in Fig. 1b, and the turbidity of the sample (which generates the broad absorption spectrum) is demonstrated in Figs. 1g and e.

Thank you for your opinion. The turbidity value is calibrated from the extinction at 500 nm. Fig. 1c describes how extinction spectra of nano-sized and micro-sized coacervates change across various wavelengths. We described Fig. 1c on Page 5.

Micro-sized coacervates exhibited a broad extinction spectrum, likely due to increased light scattering while the extinction value of nano-sized coacervates exponentially increased toward more blue-shifted wavelengths (Fig. 1c and Fig. S3).

3. Does the slight red-shift of Cy5 in Fig. 2e provide any biophysical context for the environment of the peptide in the coacervate? More discussion on this feature is needed to better understand how the peptide is incorporated within the coacervate matrix.

We used sulfo-Cy5.5, and the sulfo-Cy5.5 was linked with C5 peptides for the encapsulation.

1. Cy5.5 dye without *sulfonate group* strongly interacted with heparin, quenching its fluorescence even after the disassembly.
2. Sulfo-Cy5.5 without C5 peptide was not encapsulated within the nanocoacervates.

C5 peptide building blocks play a role in encapsulation by interacting with heparin and tyrosine peptide building blocks. We currently hypothesize the red-shifted absorption peak of sulfo-Cy5.5 arose from the increased intermolecular interactions such as pi-pi stacking²¹ between tyrosine and electrostatic interactions between heparin and lysine inside the coacervates. We now add more discussion on Page 8.

In addition, the photoluminescence (PL) performance of C7-encapsulated nano-coacervates was examined upon thrombin proteolysis. We conjugated a sulfo-Cy5.5 dye with C5 peptide using an amine-NHS coupling (*i.e.*, C7) and encapsulated C7 peptides within the nano-coacervates (further details described in Fig. S13). After C7 encapsulation, the PL signal of the C7 peptide was quenched, and the nano-coacervates exhibited a red-shifted absorption peak at 688 nm. This shift was likely due to increased intermolecular interactions such as pi-pi stacking between tyrosine²¹ and electrostatic interactions between heparin and lysine inside the nano-coacervates.

C7 peptide structure provided in Fig. S13:

Fig. S13: Sulfo-Cy5.5 conjugation to C5 peptide (i.e., C7)

a) Schematic illustration of NHS-amine coupling of sulfo-Cy5.5-NHS and C5 peptide: NH₂-YRLVPRGSYR-CONH₂. MALDI-TOF data before (b) and after (c) sulfo-Cy5.5 conjugation, confirming that sulfo-Cy5.5 dyes were successfully conjugated to C5 peptide. Sulfo-Cy5.5 needs coupling with C5 peptide to be encapsulated within the nano-coacervates. Without conjugation with the C5 peptide, sulfo-Cy5.5 was not incorporated within the nano-coacervates. A negatively charged sulfonate group on Cy5.5 is required to activate the fluorescent signal upon disassembly of the nano-coacervates. Without the sulfonate group, the positively charged Cy5.5 dyes strongly interacted with heparin, quenching its fluorescence even after disassembly. Note that sulfo-Cy5.5 without C5 peptide did not encapsulate into the nano-coacervates.

4. Controls are missing from the anti-coagulant studies presented. For example, free peptide is missing from assays presented in Fig. 2j

We now include control aPTT experiments including free peptide and tannic acid in Fig. S18 and have mentioned on Page 9. Both free C5 peptide and tannic acid show no anticoagulant effects.

In addition, the released heparin could prevent plasma coagulation as confirmed by an activated partial thromboplastin time (aPTT) test (Fig. 2j). The C5 peptide only showed plasma coagulation due to lack of anticoagulant ability (Fig S18).

Fig. S18: aPTT test of C5 peptide and tannic acid
 Plasma coagulation was observed in aPTT tests using C5 peptide and tannic acid with different concentrations from 0.1 to 1 mM.

and Fig. 4g. Similarly, free TA is required as a control in studies presented in Fig. 4g. It is possible that the non-heparin constituents of the coacervate can elicit anti-coagulant activity.

We further tested non-responsive control (scramble peptides), peptide, tannic acid, and free heparin with different concentrations. We collected fresh blood using an EDTA-treated blood collection tube. Then, we added calcium chloride to trigger blood coagulation for the test. First, different heparin concentrations were used to check anticoagulation activity as described in Fig. S35.

Fig. S35: Standard curve and F1+2 peptide concentration in human plasma
 a) A standard curve of prothrombin fragment F1+2 concentration. Activated optical density (O.D.) at 450 nm was converted to F1+2 concentrations using an equation

from the fitting curve. b) F1+2 fragment concentrations from whole human blood at different heparin concentrations from 0.2 to 1 U/ml., showing anticoagulation activity of free heparin. Whole human blood was collected using an EDTA-treated blood collection tube. Calcium chloride was used to trigger blood coagulation. 0.4 mL of whole human blood was incubated with different heparin concentrations. Low heparin concentration (lower than 0.1 U/ml) induced the strong thrombus, increasing F1+2 fragment concentrations due to thrombin activation.

After confirming the anticoagulation activity of free heparin, we then created NC-TA_{0.13} made of C6 (scramble) and C5 (thrombin-active) peptides, respectively. Then, we incubated heparin only, NC-TA_{0.13}, scramble NC-TA_{0.13}, tannic acid, and peptide only with whole human blood at the same concentrations (0.6 U/ml) for 15 min to examine blood coagulation. *Scramble NC-TA_{0.13}, TA, and C6 peptide only showed strong thrombus while heparin and NC-TA_{0.13} prevented blood coagulation as shown in Fig. 4i.* We now include all non-responsive controls on Page 15.

We tested the anticoagulant performance of NC-TA_{0.13} in whole human blood using a human prothrombin fragment 1+2 (F1+2) enzyme-linked immunosorbent assay (ELISA) kit. Fresh whole blood was collected using an EDTA-treated blood collection tube. Free heparin (0.6 U/ml), C6 peptide, TA, NC-TA_{0.13}, and scramble NC-TA_{0.13} were incubated with whole human blood at the same concentration, respectively. Calcium chloride was then used to trigger blood clot formation. We observed a negligible difference in F1+2 formation between heparin and NC-TA_{0.13}, confirming active blood anticoagulation driven by heparin released from NC-TA_{0.13} (Fig. 4i and Fig. S35-S36). In contrast, a strong thrombus was observed in TA, C5 peptide only, and scramble NC-TA_{0.13}, respectively. Note that scramble NC-TA_{0.13} was comprised of a scramble C6 peptide, serving as a non-responsive control.

Prothrombin F1+2 peptide concentrations of NC-TA_{0.13} TA, C5 peptide, and NC-TA_{0.13} made of scramble peptide (*i.e.*, C6) from whole human blood incubation. The inserted photo shows a strong blood clot from scramble NC-TA_{0.13} (right) and blood anticoagulation from NC-TA_{0.13} (left).

5. None of the results provided support a mechanism of TA encapsulation/incorporation within the coacervate bulk and could be readily explained by coating of the coacervate surface with the polyphenol. For example, the addition of a TA surface coating to the coacervates would marginally increase their hydrodynamic size (expected to be ~1nm or less). Consequently, DLS measurements are not sufficient to deconvolute encapsulation vs. surface interactions. Similarly, UV-Vis (Fig. 3c), FTIR (Fig. 3d), TEM-HAADF and EDX (Fig. e – g), and tomography (Fig. 3i) do not provide any additional clarification on encapsulation vs. surface interactions. In fact, it is unclear the purpose of TEM-HAADF and EDX given that the peptide, heparin and TA have similar elemental constituents, and therefore no reasonable way to deconvolute their spatial distribution in the coacervate. This is an important flaw in the studies given that the central innovation of the work was to avoid requiring a surface membrane/coating to stabilize the coacervate.

To investigate whether TA is distributed in the interior of the droplet, we conjugated coumarin boronic acid with TA and then encapsulated TA-coumarin conjugates inside micro-sized coacervates for confocal imaging. We confirmed the polyphenol-coumarin conjugates using a MALDI-TOF, and HPLC was used to purify free dyes before encapsulation (Fig. S228). The confocal image (Fig. 3i) shows the uniformly distributed fluorescent signal arising from inside coacervates, indicating TA encapsulation rather than membrane coating. We now describe all the information on Page 12.

Lastly, coumarin boronic acid was conjugated with TA for confocal imaging to verify TA encapsulation within coacervate droplets. HPLC was utilized to remove free coumarin dyes from TA-coumarin before encapsulation (Fig. S28). Fig. 3i shows a uniformly distributed fluorescent signal of TA-coumarin conjugates from inside the MC-TAs. This result indicates that TA is encapsulated within the coacervates rather than being membrane-coated.^{8,9}

I) confocal image of micro-coacervates encapsulating TA-coumarin conjugates. The yellow box highlights the evenly distributed fluorescent signal of TA-coumarin inside a single MC-TA. Coumarin boronic acid was linked with hydroxyl groups in TA, forming a boronate ester. The conjugates were purified using HPLC before encapsulation.

[REDACTED]

The confocal image on the right is our coacervates encapsulating TA-coumarin conjugates that show a uniformly distributed fluorescent signal. In contrast, the image on the left (from *ACS Nano* 2023, 17, 17, 16980–16992) is coacervates coated with DPPS-Au/Ag

clusters on the membrane, showing a ring.

This new finding is now available in Fig. 3 on Pages 12-13, and we cite two relevant papers that show membrane coating (showing a ring) on the coacervates for comparison. We now describe all the information on Page 12.

6. A clear translational vision of the technology was not presented. Is the design intended to enable clot-specific release of encapsulated heparin? If this is the intended clinical context, rapid heparin release preferentially at the embolism site is desired. However, heparin release was found to be significantly hindered by incorporation of the TA (which in turn is needed for stability), and required ≥ 1 hour for complete heparin release. This raises significant concerns about the actual utility of the platform.

Good point. We now include monitoring the disassembly process using heparin-FITC. We found that the addition of thrombin promptly increased the disassembly rate. PL intensity of heparin-FITC turned on within 10 min as shown in Fig. 4g. This direct heparin monitoring highlights the strength of our coacervate concepts and mitigates concern regarding the need for a full 1h activation for heparin. We now include all the information on Page 15 and experimental details on Page 20.

Both pristine nano-coacervates and NC-TA_{0.13} containing C7 peptides and heparin-FITC were incubated in 50% human plasma, respectively. The quenched fluorescence of the C7 peptide and heparin-FITC was activated as a function of the disassembly of nano-coacervate (Fig. 4e). NC-TA_{0.13} exhibited a 3.3-fold decrease in the PL activation rate of C7 peptide than NC-TA₀, indicating enhanced stability

in human plasma. Simultaneously, thrombin could accelerate heparin release from NC-TA_{0.13} (Fig. 4f). The disassembly rate of NC-TA_{0.13} increased by 4.2-fold upon the addition of thrombin (500 nM) which falls within the physiologic range of free thrombin concentration. Physiologic concentrations of free thrombin during coagulation reactions range over 500 nM.¹⁰ We also monitored this disassembly process using heparin-FITC (Fig. S32). NC-TAs exhibited a decrease in PL activation of heparin-FITC compared to pristine nano-coacervates in human plasma. Concurrently, the addition of thrombin rapidly increased the PL activation rate of heparin-FITC by 1.8-fold, recovering PL intensity within 10 min (Fig. 4g). There was no fluorescence quenching of C7 peptides and heparin-FITC either by the background medium (*i.e.*, human plasma) or by TA molecules (Fig. S33).

e) Schematic illustration of monitoring C7 peptide (f) and heparin-FITC (g) during disassembly of NC-TAs by thrombin. The left panels in (f-g) show a decrease in the PL activation rate of NC-TAs compared to nano-coacervates (*i.e.*, NC-TA₀) due to improved stability in 50% human plasma. The right panels in (f-g) illustrate the addition of thrombin rapidly activates the PL intensity of C7 peptide or heparin-FITC, indicating proteolysis-driven heparin release.

Related to the above point, experiments employing an *in vitro/ex vivo* thrombosis model that recapitulates the various physiologic factors of clotting (*e.g.*, flow, presence of complement and clotting factors, etc.) is needed to demonstrate the pre-clinical utility of the platform.

We are sorry that we did not include the thrombosis model in vitro at the current stage. However, we showed an ex vivo thrombosis model using non-responsive controls in Fig. 4i. NC-TA_{0.13} made of C6 (scramble sequence) showed strong blood coagulation while NC-TA_{0.13} (thrombin-active) prevented blood coagulation. NC-TA_{0.13} made of C5 peptide prevented blood coagulation due to rapid heparin release from thrombin cleavage while NC-TA_{0.13} made of scramble sequence exhibited strong coagulation effects. This data strongly supports the high potential of polyphenol-based coacervate stabilization and drug delivery. We now update this information on Page 15.

i) Prothrombin F1+2 peptide concentrations of NC-TA_{0.13} TA, C5 peptide, and NC-TA_{0.13} made of scramble peptide (*i.e.*, C6) from whole human blood incubation. The inserted photo shows a strong blood clot from scramble NC-TA_{0.13} (right) and the anti-coagulation from NC-TA_{0.13} (left).

We also add a discussion about the future direction of using these coacervates for future pre-clinical utility platforms. Polyphenol-encapsulated coacervates show superior coating ability derived from phenolic structures. We envision that TA-stabilized coacervates can be incorporated into medical devices such as drug-eluting stents. Future work will involve coating coacervates and demonstrating their value for cardiovascular interventions in a thrombosis model in vitro.

Coacervates containing catechol as structural building blocks have shown significant potential in drug delivery systems, particularly for gastrointestinal diseases,¹¹ due to their strong adhesiveness¹² capable of prolonged retention in the gastrointestinal tract. Likewise, our nano-coacervates strengthened by polyphenols also exhibit superior coating ability on inert substrates and maintain high stability in whole human blood (Fig. S38). Future work will involve incorporating coacervates on medical devices such as a drug-eluting stent for effective anticoagulant delivery for cardiovascular interventions.

We demonstrated the coating ability of our coacervates NC-TA_{0.13} and these coated NC-TA_{0.13} showed high stability in whole human blood (100%) for 1h as shown in Fig. S38. We now add all the information on Page 17.

Fig. S38: Optical images of MC-TAs before and after blood incubation. TA-encapsulated micro-coacervates (MC-TAs) were coated on the glass slides. a) Optical images show the MC-TAs, maintaining spherical shapes on the glass slides after drying and incubating in PBS and DMEM for 1 h. These MC-TAs showed exceptional stability after incubating in whole human blood (100%) (b). The red-dotted circle represents a single MC-TA before and after the incubation. The scale bar represents 50 μm.

8. The purpose of testing coacervate stability in non-physiologically relevant acetone, methanol, and concentrated citric acid solvents is not articulated.

Thank you for your points. We agreed that acetone, methanol, and citric solvents are physiologically not relevant. We originally included them as extremely harsh solvents to understand the boundary conditions of the coacervates. We conducted a stability test with four new biomolecules which include glutamine, glucose, human albumin, and fibrinogen which are abundant components in human blood and more physiologically relevant. The concentrations of glucose²², albumin²³, and fibrinogen²⁴ fall in physiological conditions. We observed that NC-TA_{0.13} exhibited high stability in those conditions. We added all the information in Fig. 4d and described it on Page 16, and experimental details are available on Page 20.

After identifying a critical TA encapsulation point for preserving thrombin proteolytic activity, we examined the colloidal stability of NC-TA_{0.13} under various biological environments. The NC-TA_{0.13} exhibited remarkable colloidal stability in glutamine, glucose (5.6 mM), human albumin (0.6 mM), DPBS, NaOH (pH 10), 60 C°,

NaCl (150 mM), fibrinogen (8.8 μ M), 50% of Dulbecco's Modified Eagle Medium (DMEM), serum, saliva, and urine (Fig. 4d and Fig. S30-31).

d) Size profiles of NC-TA_{0.13} in different biological environments.

9. HEK293 cells are not a suitable model to study off-target endothelial toxicity. An actual endothelial cell line (e.g., HUVEC) should be employed to look at potential cardiovascular toxicity.

We now add a new cell line (e.g., HUVEC) as a model and test particle inflammation potential and cell viability using DCF-DA and viability assays. HUVECs exhibited high cell viability and minimal ROS intensity when exposed to NC-TAs at 0.16 mM concentration. We also test its structural components including TA, heparin, and C5 peptide. All the information is now available in Fig. 4h and described on Page 15. Detailed experimental methods are available on Page 20.

We next examined the cell viability and cellular reactive oxygen species (ROS) levels of NC-TA_{0.13} using human umbilical vein endothelial cells (HUVECs), respectively. NC-TAs and their structural components such as C5 peptide, heparin, and TA showed high cell viability (>83%) and minimal ROS intensities (Fig. 4h). Cell lysis buffer was used as a positive control for cell viability assay. NC-TA_{0.13} also exhibited low cytotoxicity against human embryonic kidney (HEK) 293 cells. NC-TAs led to minimal red fluorescence of propidium iodide (PI), which corresponds to cell viability of 95% like HUVECs (Fig. S34).

h) Cell viability and ROS intensity of HUVEC incubating with PBS, TA, heparin, C5 peptide, and NC-TAs, respectively.

Reference

- 1 Lin, K. Y., Kwong, G. A., Warren, A. D., Wood, D. K. & Bhatia, S. N. Nanoparticles that sense thrombin activity as synthetic urinary biomarkers of thrombosis. *ACS nano* **7**, 9001-9009 (2013).
- 2 Capila, I. & Linhardt, R. J. Heparin–protein interactions. *Angewandte Chemie International Edition* **41**, 390-412 (2002).
- 3 Wang, J. *et al.* A mechanistic investigation of methylene blue and heparin interactions and their photoacoustic enhancement. *Bioconjugate chemistry* **29**, 3768-3775 (2018).
- 4 Yim, W. *et al.* Enhanced photoacoustic detection of heparin in whole blood via melanin nanocapsules carrying molecular agents. *ACS nano* **16**, 683-693 (2021).
- 5 Jin, Z. *et al.* Peptide Amphiphile Mediated Co-assembly for Nanoplasmonic Sensing. *Angewandte Chemie* **135**, e202214394 (2023).
- 6 Wu, D. *et al.* Phenolic-enabled nanotechnology: Versatile particle engineering for biomedicine. *Chemical Society Reviews* **50**, 4432-4483 (2021).
- 7 Pu, W., Zhao, H., Huang, C., Wu, L. & Xu, D. Visual detection of arginine based on the unique guanidino group-induced aggregation of gold nanoparticles. *Analytica chimica acta* **764**, 78-83 (2013).
- 8 Jin, Z. *et al.* Endoproteolysis of Oligopeptide-Based Coacervates for Enzymatic Modeling. *ACS nano* (2023).
- 9 Jiang, L. *et al.* Peptide-Based Coacervate Protocells with Cytoprotective Metal–Phenolic Network Membranes. *Journal of the American Chemical Society* **145**, 24108-24115 (2023).
- 10 Allen, G. A. *et al.* Impact of procoagulant concentration on rate, peak and total thrombin generation in a model system. *Journal of Thrombosis and Haemostasis* **2**, 402-413 (2004).
- 11 Zhao, P. *et al.* Nanoparticle-assembled bioadhesive coacervate coating with prolonged gastrointestinal retention for inflammatory bowel disease therapy. *Nature Communications* **12**, 7162 (2021).
- 12 Guo, Q. *et al.* Hydrogen-bonds mediate liquid-liquid phase separation of mussel derived adhesive peptides. *Nature Communications* **13**, 5771 (2022).
- 13 Di Cera, E. Thrombin. *Molecular aspects of medicine* **29**, 203-254 (2008).
- 14 Stump, D. A., Rogers, A. T., Hammon, J. W. & Newman, S. P. Cerebral emboli and cognitive outcome after cardiac surgery. *Journal of cardiothoracic and vascular anesthesia* **10**, 113-119 (1996).
- 15 Whitlock, R., Crowther, M. A. & Ng, H. J. Bleeding in cardiac surgery: its prevention and treatment—an evidence-based review. *Critical care clinics* **21**, 589-610 (2005).
- 16 Li, Y. *et al.* Molecular design principles of Lysine-DOPA wet adhesion. *Nature communications* **11**, 3895 (2020).

- 17 Lee, H., Dellatore, S. M., Miller, W. M. & Messersmith, P. B. Mussel-inspired surface chemistry for multifunctional coatings. *science* **318**, 426-430 (2007).
- 18 Berger, O. *et al.* Mussel adhesive-inspired proteomimetic polymer. *Journal of the American Chemical Society* **144**, 4383-4392 (2022).
- 19 Yim, W. *et al.* 3D-Bioprinted Phantom with Human Skin Phototypes for Biomedical Optics. *Advanced Materials* **35**, 2206385 (2023).
- 20 Olson, S. T., Halvorson, H. & Björk, I. Quantitative characterization of the thrombin-heparin interaction. Discrimination between specific and nonspecific binding models. *Journal of Biological Chemistry* **266**, 6342-6352 (1991).
- 21 McGaughey, G. B., Gagné, M. & Rappé, A. K. π -stacking interactions: alive and well in proteins. *Journal of Biological Chemistry* **273**, 15458-15463 (1998).
- 22 Borg Andersson, A., Birkhed, D., Berntorp, K., Lindgärde, F. & Matsson, L. Glucose concentration in. *Eur J Oral Sci* **106**, 931-937 (1998).
- 23 Guthrie Jr, R. D. & Hines Jr, C. Use of Intravenous Albumin in the Critically Ill Patient. *American Journal of Gastroenterology (Springer Nature)* **86** (1991).
- 24 Schlimp, C. J. *et al.* Rapid measurement of fibrinogen concentration in whole blood using a steel ball coagulometer. *Journal of Trauma and Acute Care Surgery* **78**, 830-836 (2015).

REVIEWER COMMENTS

Reviewer #1 (Remarks to the Author):

This manuscript has been revised according to previous comments, and it is suggested for publication without change.

Reviewer #2 (Remarks to the Author):

With the revised manuscript, the authors provide additional data in response to the reviewer's requests.

Among others, they characterize the tannic acid stabilization of the coacervates, the thrombin activity in plasma and serum of their experiments, and the release kinetics of heparin parallel to the peptide. They include non-responsive controls in the coagulation assays and an additional cell line for the cytotoxicity assays. The data are presented and discussed appropriately in the text.

However, the inflammatory aspects and elimination of particles from the circulation by phagocyte uptake, or clearance in kidney, spleen or liver are not addressed. These points should be handled at least with notes in the discussion.

Fig S33 f,g : Why is the release kinetics of heparin so different from the release of the peptide?

Reviewer #3 (Remarks to the Author):

The revised manuscript has thoroughly addressed the majority of prior concerns raised by this reviewer. The revised data now supports many of the key conclusions of the manuscript, and the technology is considered likely to make a lasting and significant impact to the field of cardiovascular nanomaterials.

Reviewer #1 (Remarks to the Author):

This manuscript has been revised according to previous comments, and it is suggested for publication without change.

We appreciate the reviewers for the positive compliment.

Reviewer #2 (Remarks to the Author):

With the revised manuscript, the authors provide additional data in response to the reviewer's requests. Among others, they characterize the tannic acid stabilization of the coacervates, the thrombin activity in plasma and serum of their experiments, and the release kinetics of heparin parallel to the peptide. They include non-responsive controls in the coagulation assays and an additional cell line for the cytotoxicity assays. The data are presented and discussed appropriately in the text.

We thank the reviewers for the highly positive feedback.

However, the inflammatory aspects and elimination of particles from the circulation by phagocyte uptake, or clearance in kidney, spleen or liver are not addressed. These points should be handled at least with notes in the discussion.

Thank you for your suggestions. We now have added all those comments to the discussion on page 17.

Future work will incorporate coacervates on medical devices such as a drug-eluting stents for on-demand anticoagulant delivery. Studies on inflammatory aspects such as plasma viscosity, procalcitonin, and C-reactive protein levels—as well as the elimination of particles from the circulation by phagocyte update or clearance in the kidney, spleen, and liver—are needed to validate their value in translational nanomedicine.

Fig 3 f,g : Why is the release kinetics of heparin so different from the release of the peptide?

This is because the concentration of coacervate samples and fluorescent dye-conjugates (C7 peptide and heparin-FITC) were different. We used 20 μM of C7 peptide (Fig. 3f) and 150 μM of heparin-FITC (Fig. 3g), respectively, and the number of coacervates used in Fig. 3f was higher than the number in Fig. 3g, causing the kinetic differences. Note that coacervates did not encapsulate heparin-FITC and C7 peptide at the same time. An ideal experiment would use the same number of coacervates containing a comparable number of fluorescent substrates. This has been mentioned in the material and method sections. Nevertheless, the trends of tannic-acid-driven stabilization as well as thrombin-responsiveness are the key takeaway messages in these tests. We now add a sentence to the main text on Page 15 and edit caption in Fig. 4 for clarification.

The release kinetics of heparin are different than the peptide because the concentration of coacervate samples and fluorescent dye-conjugates (C7 peptide and heparin-FITC) were different; the ratio could be tuned to control kinetics.

Reviewer #3 (Remarks to the Author):

The revised manuscript has thoroughly addressed the majority of prior concerns raised by this reviewer. The revised data now supports many of the key conclusions of the manuscript, and the technology is considered likely to make a lasting and significant impact to the field of cardiovascular nanomaterials.

We appreciate the reviewers for the highly positive compliment.

REVIEWERS' COMMENTS

Reviewer #2 (Remarks to the Author):

The authors have addressed the recommendations appropriately and I can recommend publication as is.